# Naturally acquired blocking human monoclonal antibodies to *Plasmodium vivax* reticulocyte binding protein 2b

Li-Jin Chan[1,2], Anugraha Gandhirajan[3], Lenore L. Carias[3], Melanie H. Dietrich[1], Oscar Vadas [4,5], Remy Visentin[4,5], Camila T. França [1,2], Sebastien Menant [1], Dominique Soldati-Favre [4], Ivo Mueller [1,2], Christopher L. King [3,6✉] & Wai-Hong Tham [1,2✉]

*Plasmodium vivax* preferentially invades reticulocytes and recognition of these cells is mediated by *P. vivax* Reticulocyte Binding Protein 2b (PvRBP2b) binding to human Transferrin receptor 1 (TfR1) and Transferrin (Tf). Longitudinal cohort studies in Papua New Guinea, Thailand and Brazil show that PvRBP2b antibodies are correlated with protection against *P. vivax* infection and disease. Here, we isolate and characterize anti-PvRBP2b human monoclonal antibodies from two individuals in Cambodia with natural *P. vivax* infection. These antibodies bind with high affinities and map to different regions of PvRBP2b. Several human antibodies block PvRBP2b binding to reticulocytes and inhibit complex formation with human TfR1-Tf. We describe different structural mechanisms for functional inhibition, including either steric hindrance with TfR1-Tf or the reticulocyte membrane. These results show that naturally acquired human antibodies against PvRBP2b can inhibit its function which is important for *P. vivax* invasion.

[1] The Walter and Eliza Hall Institute of Medical Research, Parkville, VIC, Australia. [2] Department of Medical Biology, The University of Melbourne, Melbourne, VIC, Australia. [3] Centre for Global Health and Diseases, Case Western Reserve University, Cleveland, OH, USA. [4] Department Microbiology and Molecular Medicine, University of Geneva, Geneva, Switzerland. [5] Proteins and Peptides Core Facility, Faculty of Medicine, University of Geneva, Geneva, Switzerland. [6] Veteran Affairs Research Service, VA Medical Center, Cleveland, OH, USA. ✉email: cxk21@case.edu; tham@wehi.edu.au

*P*lasmodium vivax is the most widespread of human malarias causing serious morbidity, occasional mortality, and recurrent relapses even after a single exposure[1]. *P. vivax* preferentially invades young red blood cells called reticulocytes. The parasite adhesin *P. vivax* Reticulocyte Binding Protein 2b (PvRBP2b) binds to Transferrin Receptor 1 (TfR1) and Transferrin (Tf) on the surface of reticulocytes, which is critical for red cell invasion[2]. PvRBP2b is a member of the *P. vivax* Reticulocyte Binding Protein (PvRBP) family of parasite adhesins, which has 10 other members[3,4]. PvRBP2b is a 326 kDa protein with a signal peptide at the N terminus and putative transmembrane domain at the C terminus[5]. TfR1 mediates the uptake of iron into cells in complex with its human ligand, iron-loaded Tf[6]. On the surface of reticulocytes, TfR1 is abundantly expressed and is shed during reticulocyte maturation into normocytes[7]. TfR1 is critical for *P. vivax* invasion, as cells expressing a mutant form of TfR1 are completely refractory to *P. vivax* invasion[2]. Mouse monoclonal antibodies (mAbs) against PvRBP2b inhibit PvRBP2b binding to reticulocytes, block complex formation with TfR1–Tf and inhibit *P. vivax* invasion by ~50% in Brazilian and Thai clinical isolates[2]. TfR1 is also the entry receptor for several human pathogens including New World hemorrhagic fever arenaviruses and hepatitis C virus[8–10].

The PvRBP2b-TfR1–Tf ternary complex has been determined by cryo-electron microscopy (cryo-EM) to 3.7 Å. A TfR1 homodimer (residues 120–760) binds two molecules of Tf (residues 1–679) and two molecules of PvRBP2b (residues 168–633)[11]. PvRBP2b interacts with TfR1 through its C-terminal domain (residues 461–633) with a buried surface area of ~1271 Å$^2$, and with Tf through its N-terminal domain (residues 168–460) with a buried surface area of ~386 Å$^2$[11]. Structure-function analyses of PvRBP2b show that none of the critical residues required for TfR1 interaction overlap with the 11 field polymorphisms identified within the TfR1–Tf binding site of PvRBP2b (residues 168–633)[11,12]. The inhibitory action of PvRBP2b-mouse mAbs has been elucidated using X-ray crystal structures of 3E9, 4F7, and 6H1 binding to PvRBP2b and a small-angle X-ray scattering (SAXS) derived model of 10B12 binding to PvRBP2b[11]. None of these PvRBP2b-mouse mAbs directly bind at the TfR1–Tf interaction sites[11]. Instead, 3E9 causes steric hindrance with TfR1 binding, and 4F7, 6H1, and 10B12 binding results in steric hindrance with the reticulocyte membrane[11].

Immuno-epidemiological studies in *P. vivax* endemic areas of Papua New Guinea, Thailand, and Brazil show a strong association between individuals with high levels of antibodies to PvRBP2b and reduced risk of *P. vivax* infection and disease[13–15]. However, it is not known if these antibodies are able to inhibit PvRBP2b function. In this study, we isolate and characterize human mAbs against PvRBP2b from peripheral blood mononuclear cells obtained from Cambodians with naturally acquired immunity to *P. vivax* and high levels of PvRBP2b antibodies, to determine their role in inhibiting either reticulocyte binding or PvRBP2b interaction with recombinant TfR1–Tf. Understanding the human antibody repertoire to PvRBP2b will aid in the discovery of novel inhibitory epitopes with implications for vaccine design and novel therapies such as passive immunization with functional human mAbs.

## Results

### Isolation and characterization of naturally acquired human mAbs to PvRBP2b.
To generate PvRBP2b human mAbs, we isolated memory B cells from two Cambodian adults whose serum inhibited PvRBP2b binding to reticulocytes, suggesting the presence of functional blocking antibodies (Supplementary Fig. S1). Using a recombinant fragment of PvRBP2b

(PvRBP2b$_{161-1454}$) to form fluorescently labeled tetramers, we isolated single PvRBP2b-reactive memory B cells, which were defined as CD19$^+$CD20$^+$IgG$^+$. Subsequent IgG gene amplification and sequencing revealed several distinct clonal groups, with some clonotypes being more abundant than others indicating clonal expansion (Fig. 1a). The majority of B cells were IgG1, and two were IgG3 (Supplementary Data 1). One or two clones were selected for expression from several of the largest clonal groups from both individuals. Cognate antibody heavy-chain and light-chain pairs were cloned into a human IgG1 backbone and expressed as full-length antibodies (Supplementary Data 1). The clonal lineages of chosen human mAbs show varying degrees of somatic hypermutations (SHM) ranging from an average of 17–35 mutations (Fig. 1b). All PvRBP2b human mAbs showed reactivity against PvRBP2b$_{161-1454}$ using an enzyme-linked immunosorbent assay (ELISA) (Fig. 1c). The human mAb against tetanus toxoid C fragment (TTCF), 043038, was used as an isotype control and showed negligible detection (Fig. 1c). The human mAbs did not recognize five other PvRBP recombinant proteins, PvRBP1a, PvRBP1b, PvRBP2a, PvRBP2c, and PvRBP2p2, and recombinant protein of the homologous *P. falciparum* Reticulocyte binding-like homolog (PfRh) family member, PfRh4, demonstrating their specificity for PvRBP2b (Supplementary Fig. 1). TTCF human mAb 043038 and a human mAb against *P. vivax* Duffy Binding Protein (PvDBP), 099100, were used to determine background signal levels. All PvRBP2b human antibodies bound to PvRBP2b with >20-fold higher signal compared to 099100 and 043038.

We determined antibody binding affinities to PvRBP2b$_{161-1454}$ to be in the low nanomolar to low picomolar range as measured by biolayer interferometry (Fig. 1d and Supplementary Fig. S2). Antibody avidity was measured by ELISA using increasing concentrations of ammonium thiocyanate to determine the concentration at which antibodies would dissociate from PvRBP2b (Fig. 1e). Antibody binding in the absence of ammonium thiocyanate was assigned 100% binding. The highest avidity antibodies showed >75% binding at 2 M ammonium thiocyanate and included 237235, 243244, 281282, 335338, 340341, and 346343 (Fig. 1e(i)). The antibodies with moderate avidity showed >75% binding at 1 M ammonium thiocyanate, but <75% binding at 2 M ammonium thiocyanate and these were 239229, 241242, 250233, 253245, 254255, 260261, 267268, and 326327 (Fig. 1e(ii)). Antibodies with the lowest avidity showed <75% binding at 1 M ammonium thiocyanate and these were 251249, 252248, 256257, 258259, 262231, 273264, 277278, 279280, and 283284 (Fig. 1e(iii)). The avidity of previously published PvRBP2b inhibitory mouse mAbs 3E9, 6H1, 10B12, and non-inhibitory mouse mAb 8G7 were also measured (Fig. 1e(iv)). 3E9, 6H1, and 10B12 were in the lowest avidity group with <75% binding at 1 M ammonium thiocyanate, while 8G7 showed moderate avidity with >75% binding at 1 M ammonium thiocyanate and <75% binding at 1 M ammonium thiocyanate (Fig. 1e(iv)). All high avidity antibodies showed picomolar affinities except 237235, which showed low nanomolar affinity. There was no correlation in the moderate and low avidity antibodies to the affinity measurements.

### Naturally acquired PvRBP2b human mAbs inhibit PvRBP2b binding to reticulocytes and block complex formation with TfR1–Tf.
To determine whether PvRBP2b antibodies block PvRBP2b$_{161-1454}$ binding to TfR1 on the surface of reticulocytes, we used a well-established flow cytometry-based reticulocyte-binding assay[2]. Antibody inhibition was expressed relative to PvRBP2b binding in the absence of antibody, which was assigned to 100% binding. We show that seven PvRBP2b antibodies, 239229,

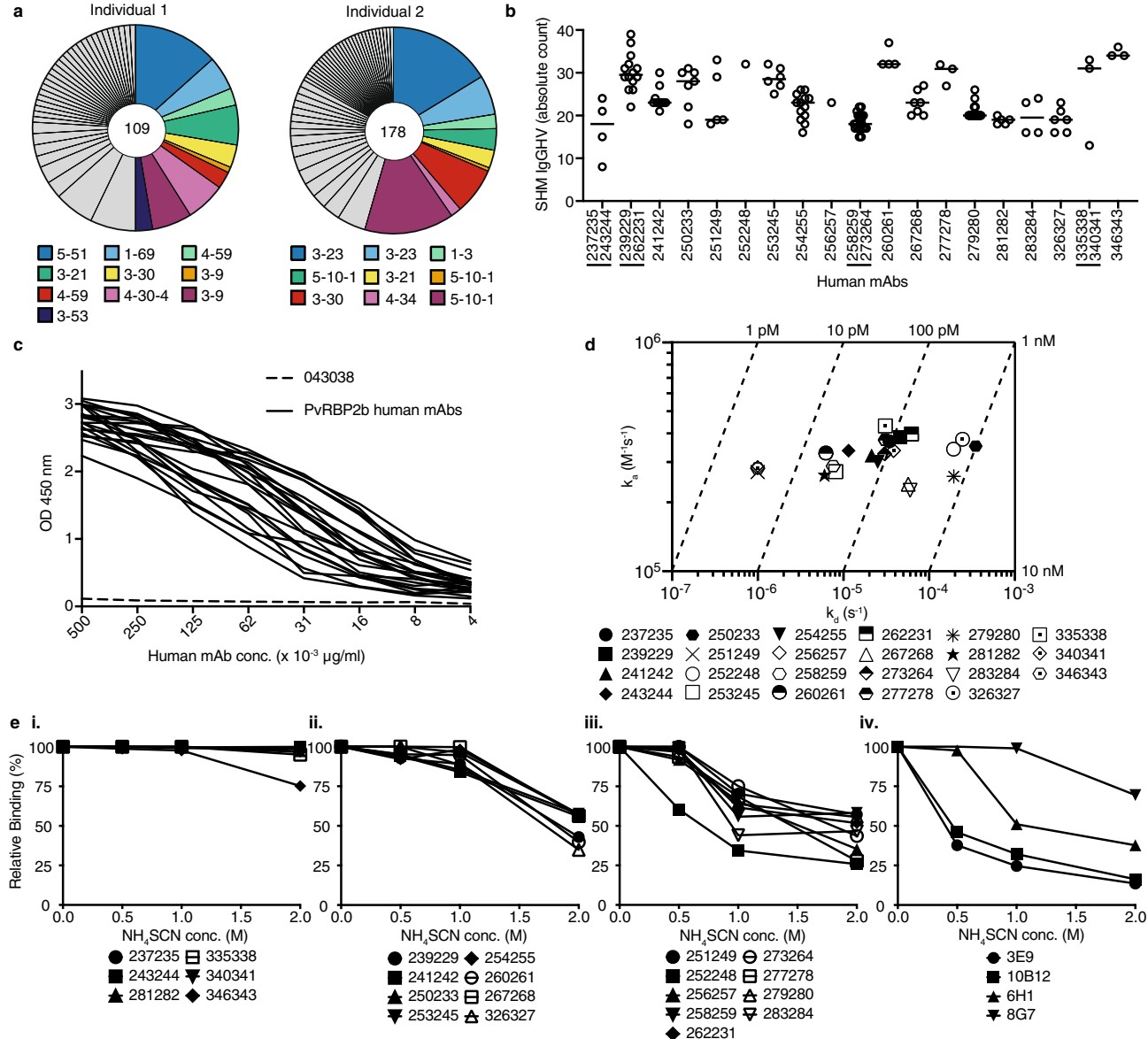

**Fig. 1 Clonal group analyses and characterization of PvRBP2b human mAbs. a** Pie charts representing the repertoire breadth of PvRBP2b$_{161-1454}$ specific single memory B cell-derived heavy-chain sequences from two Cambodian individuals. The number of variable heavy-chain gene sequences analyzed is indicated in the middle of the pie chart. The colored segments indicate the clonal groups in which at least one human mAb was expressed and their VH genes have been labeled. VH genes for clonal groups that are colored are also shown in Supplementary Data 1. **b** The frequency of somatic hypermutations (SHM) in IgG heavy-chain variable genes (IgGHV) was determined for the clonal groups of expressed human mAbs from $n = 2$ individuals. Human mAbs highlighted with a line underneath are from the same clonal group. Each circle represents a clonal member and the line represents the median number of SHM for each clonal group. **c** Reactivity against PvRBP2b$_{161-1454}$ using a dilution series of human mAbs (black solid lines). 043038 was used as isotype control (dashed line). Data are presented as mean values of two independent experiments. **d** Iso-affinity plot showing the range of dissociation rate constants ($k_d$) and association rate constants ($k_a$) of human mAbs (represented as symbols) as measured by bio-layer interferometry. Symbols that fall on the same diagonal dotted lines have the same equilibrium dissociation rate constants ($K_D$) indicated on the top and right sides of the plot. Numeric values for affinity measurements are shown in Supplementary Fig. S2 for each human mAb. **e** Avidity measurement of PvRBP2b human mAbs using increasing concentrations of ammonium thiocyanate (NH$_4$SCN) by ELISA. Human mAb binding was expressed relative to human mAb binding in the absence of NH$_4$SCN. (i) Strong binding antibodies, (ii) moderate binding, (iii) low binding, (iv) avidity of murine mAbs. Data are presented as mean values of two independent experiments. Source data are provided as a Source Data file.

241242, 253245, 258259, 262231, 273264, and 346343 inhibited PvRBP2b$_{161-1454}$ binding to reticulocytes by >98% and two PvRBP2b antibodies, 260261 and 326327, inhibited binding by 86% (Fig. 2a and Supplementary Fig. S3). The previously described PvRBP2b inhibitory mouse mAb 3E9 showed 76% binding inhibition (Fig. 2a)[2]. Nine antibodies, 237235, 243244, 251249, 252248, 254255, 256257, 281282, 335338, and 340341, inhibited PvRBP2b$_{161-1454}$ binding to reticulocytes by between 20 and 80%

and four antibodies, 250233, 267268, 277278 and 283284, showed <20% inhibition (Fig. 2a). One antibody, 279280, resulted in an increase in PvRBP2b$_{161-1454}$ binding (Fig. 2a). To control for the specificity of PvRBP2b$_{161-1454}$ inhibition, we used TTCF human mAb 043038 and PvDBP human mAb 099100[16]. Both 099100 and 043038 did not inhibit PvRBP2b$_{161-1454}$ binding (Fig. 2a).

To evaluate whether PvRBP2b human mAbs can inhibit PvRBP2b-TfR1–Tf complex formation, we utilized a fluorescence

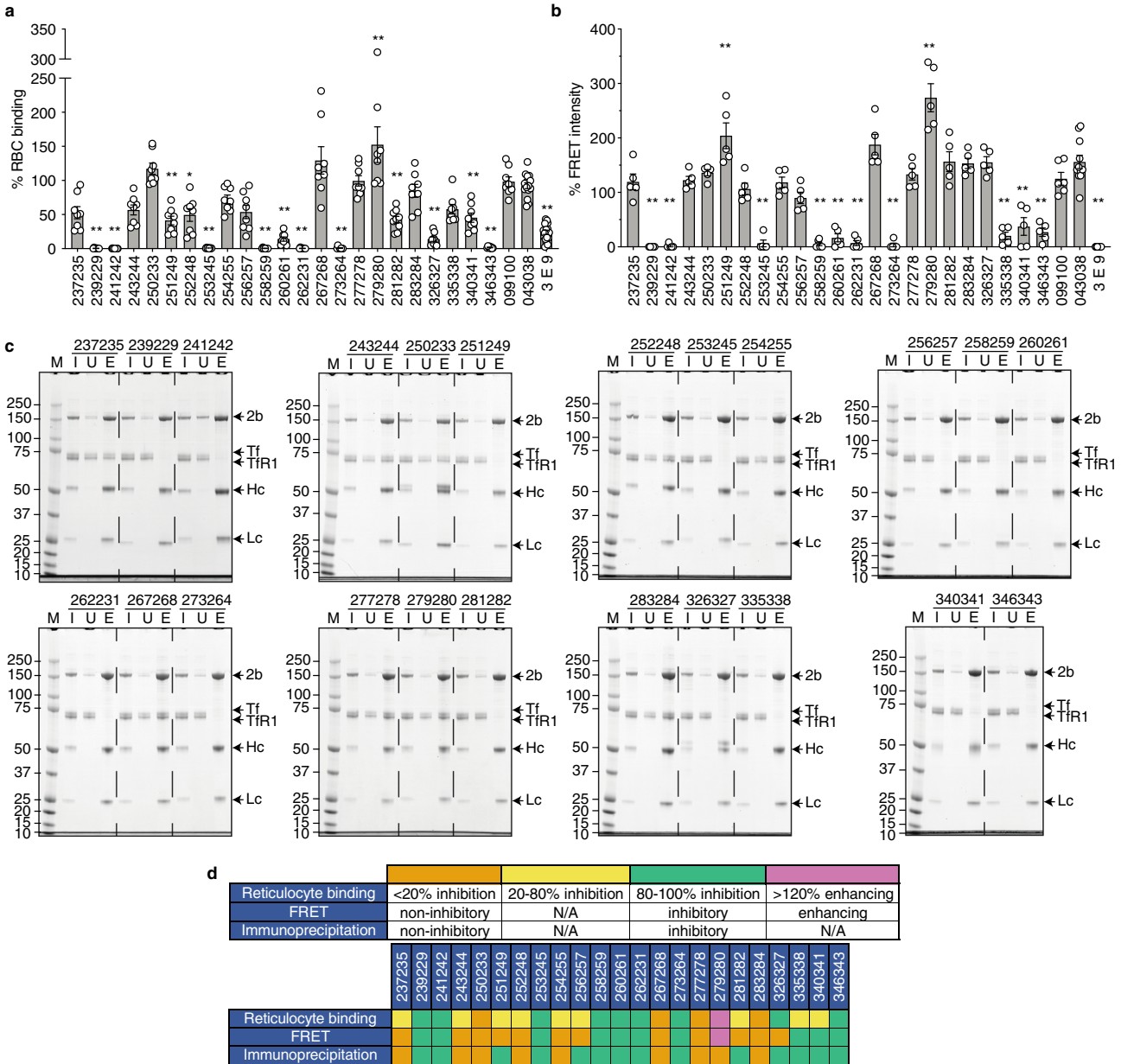

**Fig. 2 PvRBP2b human mAbs inhibit PvRBP2b$_{161-1454}$ binding to TfR1–Tf. a** PvRBP2b$_{161-1454}$ binding to reticulocytes in the presence of human mAbs was analyzed by flow cytometry. PvRBP2b$_{161-1454}$ binding in the absence of mAbs was assigned to 100%. 043038 and 099100 were used as antibody isotype controls. 3E9 is an inhibitory PvRBP2b-mouse mAb. $n = 8$ independent experiments and data are presented as mean ± SEM; circles represent independent experiments. Prism (version 8.4.3) was used to perform one-way ANOVA followed by Dunnett's multiple comparisons test using 043038 as a control to determine P-values. *$P \leq 0.001$, **$P \leq 0.0001$. **b** The ability of PvRBP2b human mAbs to inhibit PvRBP2b$_{161-1454}$-TfR1–Tf complex formation was analyzed in the FRET-based assay. The FRET intensity in the absence of mAbs was assigned to 100%. $n = 5$ independent experiments and data are presented as mean ± SEM; circles represent independent experiments. Prism (version 8.4.3) was used to perform one-way ANOVA followed by Dunnett's multiple comparisons test using 099100 as a control to determine P-values. **$P \leq 0.0001$. **c** IP of PvRBP2b$_{161-1454}$ using PvRBP2b human antibodies in the presence of TfR1 and Tf were examined by reducing SDS-PAGE. $n = 1$ independent experiment. 2b, PvRBP2b; Hc, mAb heavy chain; Lc, mAb light chain; I, input; U, unbound; E, eluate; M, molecular weight marker with molecular weights labeled on the left side of the gels in kDa. **d** Summary table of PvRBP2b human mAb phenotypes in the reticulocyte-binding, FRET, and IP assays. The different phenotypes indicated in orange, yellow, green, and pink for each assay have been shown in the legend. Source data are provided as a Source Data file.

resonance energy transfer (FRET)–based assay[2]. FRET signal in the absence of antibody was designated as 100% FRET signal. PvDBP human mAb 099100 and TTCF human mAb 043038 isotype controls increased the FRET signal by 125% and 156%, which was also observed for several PvRBP2b human mAbs, and these ranges were considered non-inhibitory (Fig. 2b). As a positive control for inhibition, we utilized PvRBP2b-mouse mAb 3E9 which has been previously shown to inhibit the complex

formation and abolish the FRET signal[2] (Fig. 2b). We observed that PvRBP2b antibodies 239229, 241242, 253245, 258259, 260261, 262231, 273264, and 346343, blocked PvRBP2b-TfR1–Tf complex formation as indicated by a > 70% decrease in FRET signal (Fig. 2b). This collection of antibodies also inhibited PvRBP2b$_{161-1454}$ binding to reticulocytes by >98%, with the exception of 260261, which showed 86% inhibition (Fig. 2a). 335338 and 340341 reduced the FRET signal by 80% and 63%

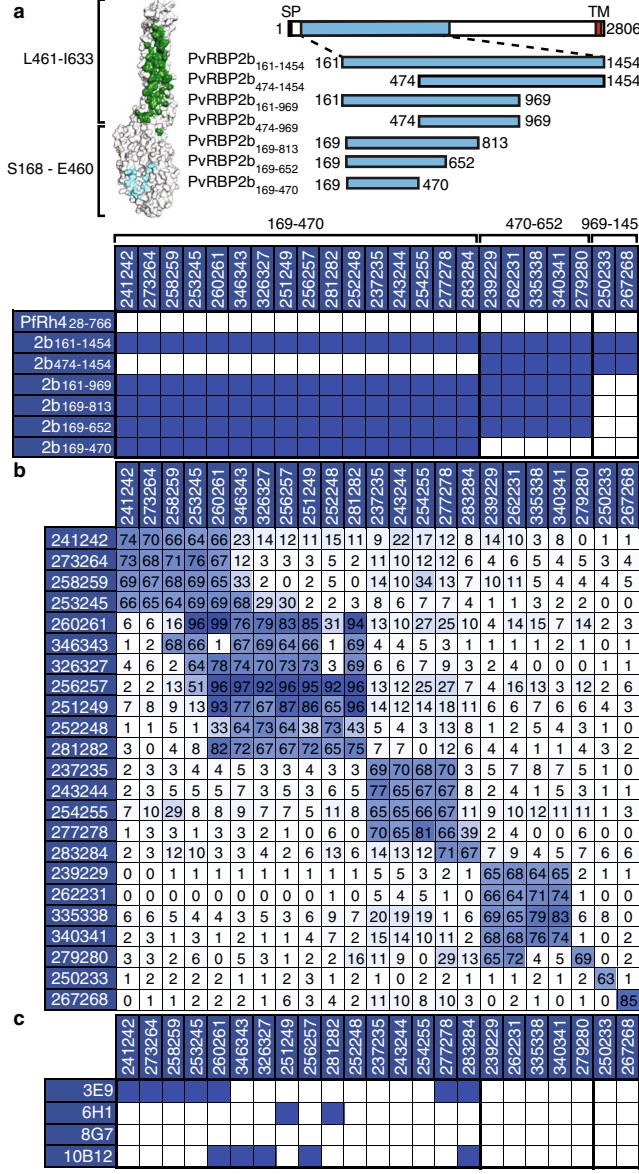

**Fig. 3 Epitope mapping of PvRBP2b human mAbs. a** (Left) Surface representation of PvRBP2b from the cryo-EM ternary complex structure (PDB accession code 6D04). PvRBP2b residues S168–E460 and L461-I633 are demarcated to show the domains that mainly interact with Tf (cyan) and TfR1 (green), respectively. (Right) Schematic representation of full-length PvRBP2b and recombinant protein fragments. Numbers indicate N- and C-terminal amino acid residues. SP, signal peptide; TM, transmembrane domain (red box). (Bottom) Mapping of PvRBP2b human mAbs using recombinant protein fragments. Blue boxes represent signals that are >10-fold higher compared to PfRh4. The amino acid region to which human mAbs bind are indicated above the table. **b** Competition ELISA using immobilized PvRBP2b human mAbs indicated on the left column incubated with a mixture of human mAbs indicated on the top row and PvRBP2b$_{161-1454}$ using a 20:1 molar ratio. Inhibition of PvRBP2b$_{161-1454}$ binding in the presence of antibody in solution was calculated relative to PvRBP2b$_{161-1454}$ binding in the absence of antibody in solution. A blue to white gradient shows antibodies with the highest levels of competition in dark blue and the lowest in white. Competition (>60% inhibition of PvRBP2b$_{161-1454}$ binding), partial competition (between 30 and 59%), no competition (<29%). **c** Competition ELISA using immobilized PvRBP2b-mouse mAbs incubated with a mixture of human mAbs and PvRBP2b$_{161-1454}$ at a 20:1 molar ratio. Blue boxes indicate competition.

(Fig. 2b), whereas they inhibited PvRBP2b$_{161-1454}$ binding to reticulocytes by 42% and 54% (Fig. 2a). 279280 increased the FRET signal by 274%, and likewise also enhanced PvRBP2b$_{161-1454}$ binding to reticulocytes (Figs. 2a and 2b). All other PvRBP2b antibodies, 237235, 243244, 250233, 251249, 252248, 254255, 256257, 267268, 277278, 281282, 283284, and 326327, showed no or <10% decrease in FRET signal (Fig. 2b). However, of the aforementioned human mAbs, 326327 has previously been shown to inhibit reticulocyte binding by 86% (Fig. 2a).

We performed co-immunoprecipitation (IP) assays using PvRBP2b human mAbs with PvRBP2b$_{161-1454}$, TfR1, and Tf to observe if the human mAbs could block complex formation. All PvRBP2b antibodies were able to immuno-precipitate recombinant PvRBP2b$_{161-1454}$ (Fig. 2c). In the presence of PvRBP2b human mAbs 239229, 241242, 251249, 253245, 256257, 258259, 260261, 262231, 273264, 281282, 326327, 335338, 340341, and 346343, the TfR1–Tf bands were absent from the eluate lanes, suggesting that these antibodies blocked TfR1–Tf binding to PvRBP2b$_{161-1454}$ (Fig. 2c). In contrast, 237235, 243244, 250233, 252248, 254255, 267268, 277278, 279280, and 283284 immuno-precipitated PvRBP2b$_{161-1454}$ in complex with TfR1 and Tf, and were therefore non-inhibitory in this assay (Fig. 2c).

Collectively, we have identified eight PvRBP2b human mAbs, 239229, 241242, 253245, 258259, 260261, 262231, 273264, and 346343, that inhibited the binding of PvRBP2b to reticulocytes and blocked complex formation in both the FRET and IP assays (Fig. 2d).

**Mapping the binding sites of PvRBP2b human mAbs using PvRBP2b fragments.** We utilized a panel of PvRBP2b recombinant fragments to determine the region bound by PvRBP2b human mAbs[2]. Our results show that PvRBP2b human mAbs bind to three different regions. The majority of PvRBP2b human mAbs, 237235, 241242, 243244, 251249, 252248, 253245, 254255, 256257, 258259, 260261, 273264, 277278, 281282, 283284, 326327, and 346343, bound epitopes within PvRBP2b$_{169-470}$ (Fig. 3a and Supplementary Fig. S4). 239229, 262231, 279280, 335338, and 340341 bound epitopes between residues 470–652, and 250233 and 267268 bound epitopes between residues 969–1454 (Fig. 3a). As expected, none of the antibodies showed reactivity to the negative control, PfRh4$_{28-766}$. (Supplementary Fig. S4). TTCF human mAb 043038 showed no reactivity against any of the PvRBP2b recombinant fragments or PfRh4$_{28-766}$ (Supplementary Fig. S4).

To evaluate whether PvRBP2b human mAbs recognized similar epitopes within PvRBP2b, antibody competition experiments were performed using ELISA plates coated with each human mAb. To identify blocking activity, biotinylated PvRBP2b$_{161-1454}$ was pre-incubated with individual PvRBP2b human mAbs in solution prior to incubation on the antibody-coated ELISA plates. No competition, partial competition, and full competition were characterized as 0–29%, 30–59%, and 60–100% inhibition of biotinylated PvRBP2b$_{161-1454}$ binding relative to PvRBP2b$_{161-1454}$ binding alone, respectively. As expected, all PvRBP2b human mAbs competed with themselves (Fig. 3b). Some PvRBP2b human mAbs showed non-reciprocal responses depending on which antibody was coated and which was in solution, but they were nonetheless considered competing antibodies (Fig. 3b). 241242, 273264, 258259, and 253245 competed with each other and were inhibitory in the reticulocyte-binding, FRET, and IP assays (Figs. 2d and 3b). 260261 and 346343 were also inhibitory across all functional assays and while 260261 competed with all the aforementioned antibodies, 346343

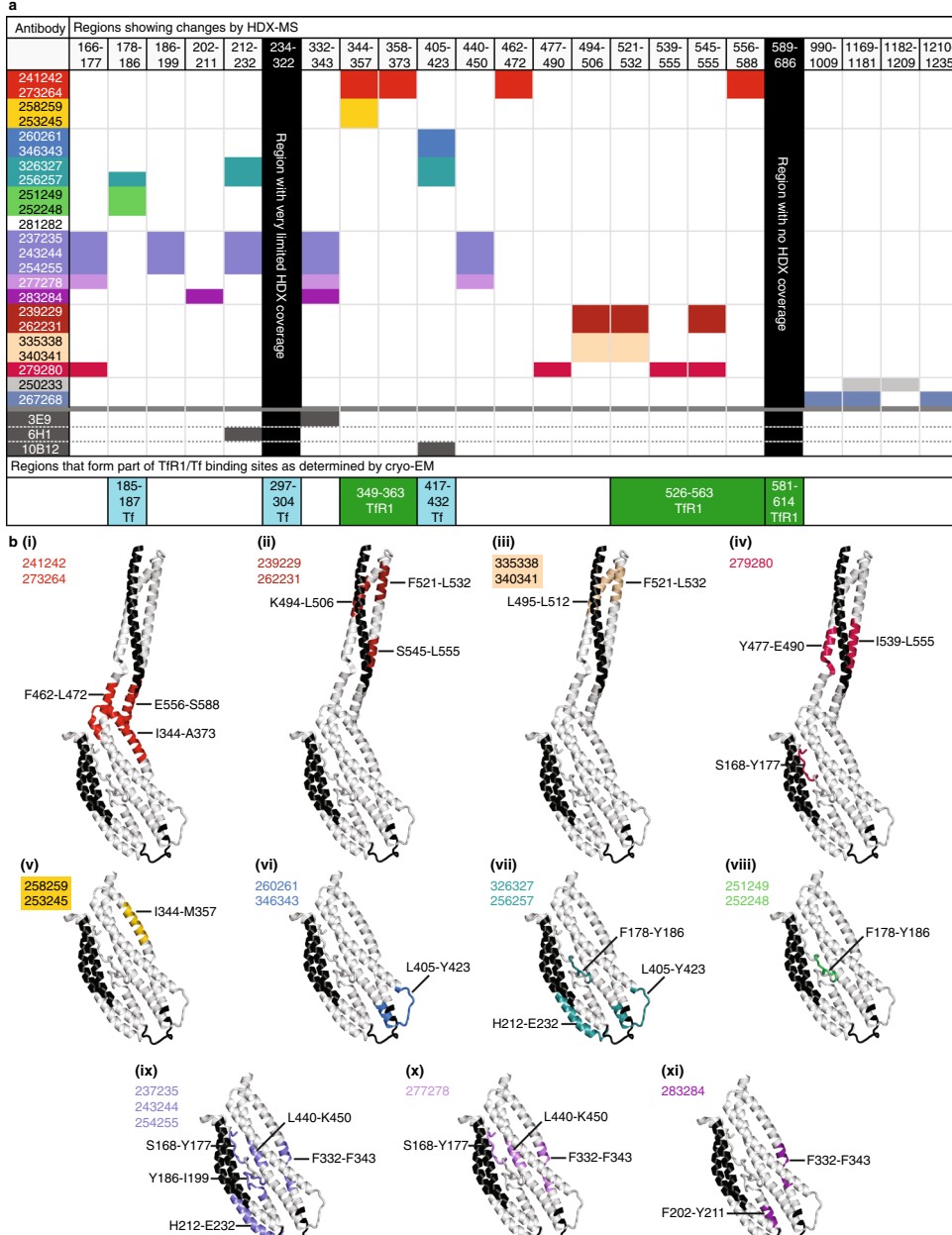

**Fig. 4 HDX-MS analysis of PvRBP2b regions affected upon PvRBP2b human mAb binding. a** Table showing regions of PvRBP2b that are protected from deuteration upon PvRBP2b human mAb binding. Antibodies that show protection in similar regions are grouped by colors. Colored boxes indicate that the antibody shows protection in the region specified in the top row. Black columns indicate regions where no or few peptides could be detected by mass spectrometry. The bottom-most row shows regions that also form part of the TfR1–Tf binding sites. **b(i)-(xi)** Mapping of PvRBP2b protected regions on X-ray crystal structures. The colored regions correspond to the table in **a** for different groups of antibodies. The protected regions for human mAbs 250233 and 267268 could not be shown structurally as the cryo-EM structure only extends to 633 residues. Black indicates regions that could not be analyzed by HDX-MS.

competed with 258259 and 253245, but not 241242 and 273264 (Figs. 2d and 3b). Both 260261 and 346343 also competed with 326327, 256257, 251249, 252248, and 281282, which all competed with each other (Fig. 3b). Antibodies 237235, 243244, 254255, and 277278 competed with each other, and all show weaker inhibition (~20–80%) in the reticulocyte-binding assay (except 277278 that showed no inhibition), and none were inhibitory in either the FRET or IP assays (Figs. 2d and 3b). 277278 also competed with 283284, which did not compete with any other antibody and was also not inhibitory in any of the interaction assays (Figs. 2d and 3b). 239229, 262231, 335338, and 340341 that

bound between residues 470–652 competed with each other (Fig. 3b). They were inhibitory across all three interaction assays, but 335338 and 340341 were only 20–80% inhibitory in the reticulocyte-binding assay (Fig. 2d). 279280 competed with 239229 and 262231, and 279280 enhanced PvRBP2b$_{161-1454}$ binding to reticulocytes and increased the signal intensity in the FRET-based assays (Figs. 2d and 3b). 250233 and 267268 that bound between residues 969–1454 were not inhibitory in any assays and did not compete with each other (Figs. 2d and 3b). In general, the competition ELISA results are consistent with the domain mapping experiments, and human mAbs that compete

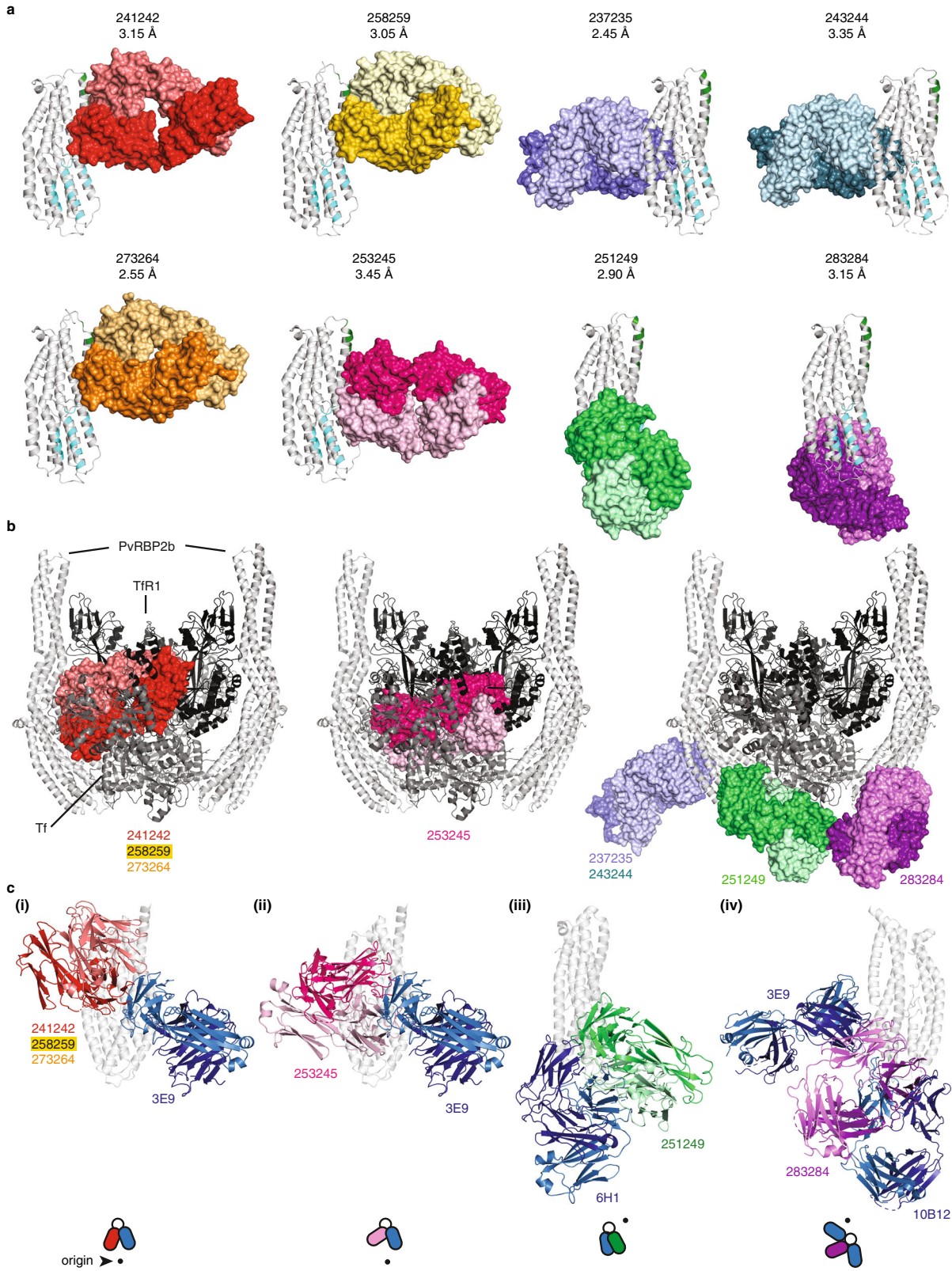

with each other share similar phenotypes as observed in the functional assays.

We wanted to determine whether PvRBP2b human mAbs bound similar epitopes compared to PvRBP2b-mouse mAbs[2]. Mouse mAbs 3E9, 6H1, and 10B12 bind PvRBP2b$_{169-470}$, block PvRBP2b$_{161-1454}$ binding to reticulocytes and inhibit ex vivo *P. vivax* invasion to ~50%[2]. 8G7 is a PvRBP2b-mouse mAb that

binds residues 813-1454 and has no effect on *P. vivax* invasion[2]. PvRBP2b human mAbs 241242, 273264, 258259, 253245, 260261, 277278, and 283284 competed with 3E9 (Fig. 3c and Supplementary Fig. S5). 260261, 283284, 346343, 326327, and 256257 competed with 10B12, whereas 251249 and 281282 competed with 6H1 (Fig. 3c). None of the 23 PvRBP2b human mAbs competed with 8G7 (Fig. 3c).

**Fig. 5 Structural analysis of PvRBP2b human mAb epitopes. a** Crystal structures of PvRBP2b$_{169-470}$ (white) with antibody Fab fragments of human mAbs 241242, 258259, 273264, 253245, 237235, 243244, 251249, and 283284. On PvRBP2b, the regions that interact with TfR1 and Tf are highlighted in green and cyan, respectively. Fabs are shown in surface representation (heavy and light chains are indicated in a dark and light color, respectively).
**b** Superimposed structures of the cryo-EM complex of PvRBP2b-TfR1–Tf (PvRBP2b, white; TfR1, black; Tf, gray) with crystal structures of human mAbs that directly interfere with TfR1 binding (left and middle panel) and human mAbs that do not interfere with TfR1 binding (right panel). 241242 (red) was shown as a representative for 258259 and 273264 structures, and 237235 (cyan) was shown as a representative for 243244 structure, as these groups bind in a similar manner to PvRBP2b. **c** Superimposed structures at different orientations showing steric hindrance of human mAbs with mouse mAbs. (i) 241242 with a crystal structure of mouse mAb 3E9, (ii) 253245 with 3E9, (iii) 251249 with a crystal structure of mouse mAb 6H1, (iv) 283284 with 3E9 and SAXS structure of 10B12. The top-view schematic at the bottom of each superimposed structure shows each view relative to **c**(i) where the origin (black dot) is the front view of **c**(i).

**Hydrogen–deuterium exchange epitope mapping**. To map antibody epitopes for PvRBP2b human mAbs in a near-native environment, we used hydrogen–deuterium exchange coupled to mass spectrometry (HDX-MS). The method identifies interaction sites by comparing protein dynamics in the absence and presence of a ligand[17]. We determined HDX rates for PvRBP2b$_{161-1454}$ alone and in complex with each human mAb. To identify the best HDX conditions that would allow us to screen all antibodies, we first tested different incubation times for PvRBP2b$_{161-1454}$ alone both at 0 °C and at room temperature (22 °C) (Supplementary Data 2). These experiments showed that incubation times of 5 min at either 0 or 22 °C were the most appropriate as they showed intermediate levels of deuterium incorporation for a majority of peptides, both for the fast-exchanging and slow-exchanging regions of the protein. PvRBP2b was very stable in all tested labeling conditions and we obtained above 85% protein coverage, with limited information for regions 234–322 and 589–686 (Supplementary Fig. S6). Identification of PvRBP2b epitopes for mouse antibodies 3E9, 6H1, and 10B12 highly correlate with the previously published X-ray and SAXS structures[11], validating our approach to identify human antibody epitopes (Supplementary Fig. S7). Some antibodies can be grouped as they show a similar pattern of HDX changes on RBP2b (Fig. 4a and Supplementary Fig. S7).

A group of four human antibodies, 241242, 273264, 258259, and 253245, all show changes in the 344-373 region, which is located in the PvRBP2b N-terminal domain (residues 166–460) next to the junction joining the N-terminal domain to the C-terminal domain (residues 461–633) (Fig. 4b(i) and (v)). Interestingly, antibodies 241424 and 273264 show additional changes in the C-terminally located regions 462–472 and 556–588 (Fig. 4b(i)). Antibodies 260261, 346343, 326327, and 256256 show HDX changes in a loop at the extremity of PvRBP2b, peptide region 405-423, an epitope that overlaps with that of the PvRBP2b inhibitory mouse mAb 10B12[11] (Fig. 4b(vi) and (vii)). Additionally, antibodies 326327 and 256256 also show changes in a second loop on the same extremity of PvRBP2b, residues 212–232 (Fig. 4b(vii)). The third loop that is sandwiched between the two loops showing HDX changes is unfortunately not amenable to analysis by HDX-MS due to the lack of high-quality peptides that cover this region (Supplementary Fig. S6). Antibodies 251249 and 252248 show HDX changes in a loop that extends from the Tf binding interface of PvRBP2b, encompassing residues 178–186 (Fig. 4b(viii)). Antibody 281282 likely binds very similarly to 251249 and 252248 as all three antibodies have the same profile in the competition assay (Fig. 3b), but no HDX changes were observed in the presence of antibody 281282 (Fig. 4a). This may be because the chosen HDX incubation time and temperatures do not permit observable changes in the region where 281282 is binding, or because the antibody is contacting a region that is not covered in the HDX-MS analysis. Antibodies 237235, 243244, and 254255 show multiple regions with HDX changes: PvRBP2b regions 166–177, 186–199, 212–232, 332–343,

and 440–450, suggesting that the binding of these antibodies affects multiple sites (Fig. 4b(ix)). The epitope of antibody 277278 partially overlaps with the first three antibodies, with regions 332–343 and 440–450 showing the most significant changes (Fig. 4b(x)). The epitope of antibody 283284 also partially overlaps with 237235, 243244, 254255, and 277278 in the region 332–343 and additionally protects region 202–211 (Fig. 4b(xi)). Of the antibodies that bind to the C-terminal domain, antibodies 239229, 262231, 335338, and 340341 protect regions 494–506 and 521–532, with antibodies 239229 and 262231 showing additional protection of the 545–555 region (Fig. 4b(ii) and (iii)). The latter region is part of the interface that binds TfR1. 279280 also shows changes in the C-terminal domain and has a dual-site epitope encompassing residues 477–490 and 539–555 (Fig. 4b(iv)). Antibodies 267268 and 250233 show HDX changes in regions distant from both the Tf and TfR1 interaction sites (Fig. 4a). None of the antibodies showed increased PvRBP2b$_{161-1454}$ HDX rates upon binding.

The data obtained by HDX-MS show an excellent correlation with the antibody competition assay and with the PvRBP2b recombinant protein fragment mapping, defining the epitope of each antibody with a much higher spatial resolution. In addition, the data are in agreement with previously obtained PvRBP2b-mouse mAbs structures obtained either by X-ray crystallography or SAXS[11].

**Structural characterization of PvRBP2b human mAbs**. To further elucidate how human mAbs recognized PvRBP2b, we structurally characterized several antibody Fab fragments in complex with PvRBP2b$_{169-470}$. We were able to determine eight crystal structures of PvRBP2b$_{169-470}$ with antibody Fabs with resolutions ranging from 2.45 to 3.45 Å—PvRBP2b-241242 (3.15 Å), PvRBP2b-258259 (3.05 Å), PvRBP2b-273264 (2.55 Å), PvRBP2b-253245 (3.45 Å), PvRBP2b-237235 (2.45 Å), PvRBP2b-243244 (3.35 Å), PvRBP2b-251249 (2.90 Å) and PvRBP2b-283284 (3.15 Å) (Fig. 5a).

241242, 258259, and 273264 form interactions with PvRBP2b$_{169-470}$ at helices α5 and α6, and 241242 forms additional interactions with part of the N-terminal loop between residues S181–N185 (Fig. 5a). These antibodies bind epitopes that partially overlap with the TfR1 binding site at helix α5 (Fig. 5a, green highlight). 253245 binds an adjacent overlapping epitope and makes contacts with α5, α6, α7, and residues S184–Y186 of the N-terminal loop. Our structural analyses that show all four antibodies binding overlapping epitopes are consistent with the results from the competition ELISA (Fig. 3b). 258259 and 273264 are from the same clonal group and as expected the crystal structures show they adopt similar heavy and light-chain CDR loop conformations upon binding PvRBP2b (Supplementary Fig. S8). Although 241242 belongs to a different clonal group, it has similar CDR loop conformations to 258259 and 273264 (Supplementary Fig. S8). Superimposing the crystal structures of PvRBP2b-241242, PvRBP2b-253245, PvRBP2b-258259, and

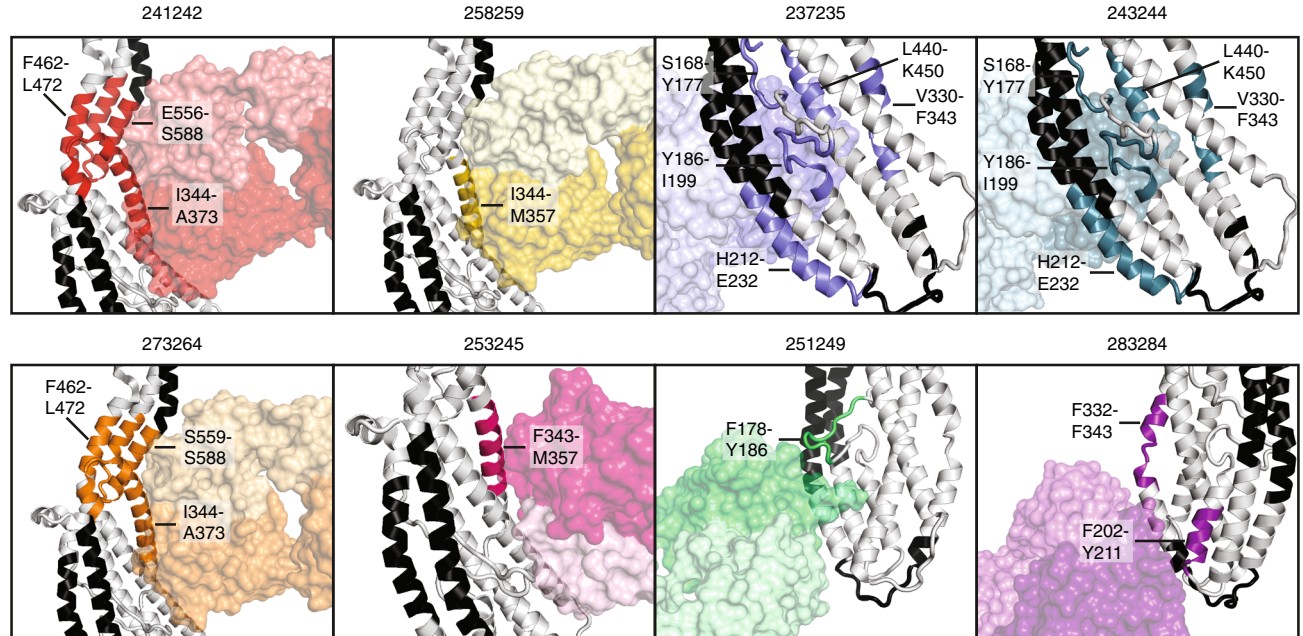

**Fig. 6 Comparison between HDX protected regions and crystal structures.** Combination of HDX-MS data with crystal structures of PvRBP2b human Fab fragments bound to PvRBP2b. Fabs are shown in transparent surface representation. Regions that show protection by HDX-MS for each mAb are colored and the residue range is labeled. Black indicates the regions where no peptides were detected by HDX-MS.

PvRBP2b-273264 onto the PvRBP2b-TfR1–Tf cryo-EM ternary complex reveals that these antibodies cause steric hindrance with TfR1–Tf (Fig. 5b). The ability of these antibodies to directly interfere with TfR1–Tf binding is consistent with these antibodies showing inhibition in all interaction assays (Fig. 2d).

251249 contacts PvRBP2b$_{169-470}$ at helices α3 and α7 and the N-terminal loop between E183 and Y187 (Fig. 5a). It binds just adjacent to the Tf binding site. PvRBP2b-251249 when super-imposed onto the PvRBP2b-TfR1–Tf complex shows steric occlusion with Tf and its heavy-chain CDR3 loop contacts the region where Tf binds PvRBP2b (Fig. 5b). 251249 is 59% inhibitory in the reticulocyte-binding assay and inhibitory in the IP assay, but not in the FRET assay and appears to act through interference with Tf binding (Fig. 2d).

237235 and 243244 contacts PvRBP2b$_{169-470}$ helices α2 and α7 and part of the N-terminal loop between residues D173–N174 and Q191–P194 (Fig. 5a). 237235 and 243244 belong to the same clonal group and they bind similarly to their target epitopes with heavy and light-chain CDR loops adopting similar conformations when bound to PvRBP2b (Supplementary Fig. S8). When superimposed onto the PvRBP2b-TfR1–Tf complex, PvRBP2b-237235, and PvRBP2b-243244 show no overlap with TfR1–Tf binding, but they do bind directly opposite the Tf binding site (Fig. 5b). While these antibodies are not inhibitory in the FRET and IP assays, they inhibit PvRBP2b binding to reticulocytes by 47% and 43%, which suggests these antibodies may affect Tf binding from the opposite face of the Tf binding site (Fig. 2d). Finally, 283284 forms interactions with helices α1 and α4 of PvRBP2b169–470. When superimposed onto the PvRBP2b-TfR1–Tf complex, 283284 shows no interference with TfR1–Tf binding (Fig. 5b). This is consistent with 283284 showing no inhibition in any of the interaction assays (Fig. 2d).

We also superimposed the crystal structures of PvRBP2b human mAbs with inhibitory mouse mAbs 3E9 and 6H1 and the SAXS model of 10B12 to determine if there was any overlap. 3E9 shows steric hindrance with 241242, 253245, 258259, 273264, and 283284, while 6H1 shows steric hindrance with 251249 (Fig. 5c). Interestingly, 3E9 and 10B12 shows steric hindrance

with 283284 (Fig. 5c). These superimposed structures are consistent with the competition ELISA results between the human and mouse mAbs (Fig. 3c).

The antibody epitopes revealed by X-ray crystal structures corroborate regions that show HDX-MS changes (Fig. 6). HDX-MS also reveals changes in PvRBP2b regions that do not directly form antibody interactions as shown in the crystal structures. In particular, 237235 and 243244 that interact with N-terminal loop regions D173–N174 and Q191–P194 also show HDX-MS changes in the loop between Y186 and Q191, a region that forms interactions with Tf (Fig. 6).

## Discussion

Longitudinal cohort studies in Papua New Guinea, Thailand, and Brazil show that individuals with higher levels of antibodies to PvRBP2b are associated with a reduced risk of *P. vivax* infection and disease[14,15,18]. To understand the putative protective function of human PvRBP2b antibodies, we isolated and characterized mAbs to PvRBP2b from individuals with naturally acquired immunity to *P. vivax*. We show that natural infection elicits PvRBP2b-specific human mAbs that inhibit PvRBP2b binding to reticulocytes and block complex formation between PvRBP2b and TfR1–Tf. X-ray crystallography combined with HDX-MS reveals the epitopes of 22 PvRBP2b antibodies and these results show how naturally acquired human mAbs block PvRBP2b function.

Out of the 23 PvRBP2b human mAbs isolated and examined in detail, eight of these including 239229, 241242, 253245, 258259, 260261, 262231, 273264, and 346343 were inhibitory across all three of our functional assays; reticulocyte-binding, FRET, and IP (Fig. 2d). Crystal structures of PvRBP2b-241242, PvRBP2b-253245, PvRBP2b-258259, and PvRBP2b-273264 show that these antibodies bind to the PvRBP2b-TfR1 interaction interface, and therefore inhibit through steric hindrance with TfR1 (Fig. 5b). While we were unsuccessful in our crystallography efforts with 260261 and 346343, HDX-MS data show that the direct interface lies at the Tf interaction site (Fig. 4b(iii)). These antibodies also show inhibition across all three interaction assays, which strongly suggests that they cause direct steric hindrance with TfR1–Tf

(Fig. 2d). The inhibitory antibodies 239229 and 262231 show amide-exchange protection of regions K494-L512, F521-L532, and S545-L555, which harbors the extensive interaction site for TfR1 binding (Fig. 4)[11]. Unfortunately, this domain has been recalcitrant to our crystallization efforts. However, we hypothesize that these antibodies directly interfere with TfR1 binding due to their ability to inhibit across all three interaction assays (Fig. 2d).

The PvRBP2b-251249 crystal structure shows that this antibody binds adjacent to the Tf interaction site (Fig. 5b). PvRBP2b-251249 shows steric hindrance with Tf when superimposed onto the ternary complex. HDX-MS shows protection of the loop between residues F178-Y186, which is part of the interface with Tf (Figs. 4b(v) and 5b). Therefore, 251249 may destabilize the interaction between PvRBP2b and Tf. Disruption of Tf binding may result in an intermediate inhibitory phenotype as 251249 shows 59% inhibition in the reticulocyte-binding assay compared to the >98% inhibition observed with human mAbs that sterically hinder TfR1 binding (Fig. 2a). In competition assays, 251249 competes with human mAbs 252248 and 256257, which like 251249 show protection in HDX-MS between residues 178–186 (Figs. 3b and 4a). These antibodies show 51% and 46% inhibition in the reticulocyte-binding assay respectively, therefore they may also inhibit complex formation by disrupting PvRBP2b binding to Tf (Fig. 2a).

Based on our HDX-MS results, we propose that the binding of antibodies 237235, 243244, and 254255 results in allosteric changes in the PvRBP2b regions interacting with Tf (Figs. 4 and 6). Crystal structures of 237235 and 243244 show they bind PvRBP2b on the opposite side of the TfR1-Tf binding site (Fig. 5b). Despite this, they are able to inhibit PvRBP2b$_{161-1454}$ binding by 47% and 43% in the reticulocyte-binding assay (Fig. 2a). HDX-MS revealed that when these antibodies bind they cause additional changes in part of a loop (Y186-I199) that forms the interface with Tf (Fig. 4). Antibodies 250233, 267268, 277278, and 283284, which showed no inhibition in the reticulocyte-binding assay (Fig. 2a), have partially overlapping epitopes with 237235 and 243244 but showed no observable changes in the region between Y186-I199 by HDX-MS (Fig. 4). It is unclear how changes in this loop may disrupt PvRBP2b binding. Previous studies have shown that Y186 of PvRBP2b is the only residue in this loop that forms a hydrogen bond with E265 of Tf, but mutation of Y186 to alanine does not cause a defect in PvRBP2b binding[11]. One possibility is that the overall conformation or position of the loop is important for Tf binding.

326327 is the only human mAb that is inhibitory in the reticulocyte-binding assay and IP assay, but not the FRET-based assay (Figs. 2d and 3c). The mode of inhibition for 326327 may be inferred by shared characteristics with previously characterized PvRBP2b-mouse mAb 10B12[11]. 10B12 is also inhibitory in the reticulocyte-binding and IP assays but shows no inhibition in the FRET-based assay[2]. 326327 competed with 10B12 for PvRBP2b binding, which is validated by HDX-MS analysis identifying a rather extended epitope at the extremity of PvRBP2b, overlapping with 10B12, 256257, 260261, and 346343 (Figs. 3c and 4). 326327 binding also likely triggers allosteric changes in regions where other antibodies bind (6H1, 256257, 260261, and 346343). The SAXS solution structure of the PvRBP2b-10B12 complex shows that it does not bind to the TfR1-Tf interaction interface, but may instead cause steric hindrance with the reticulocyte membrane[11]. We propose that 326327 binds an epitope overlapping with 10B12 that results in steric hindrance with the reticulocyte membrane. Antibodies are able to cause steric hindrance with the reticulocyte membrane as TfR1 is tethered to membranes by a 2.9 nm molecular stalk that joins to a single-pass transmembrane domain[19]. Antibodies binding in the right orientation could

therefore prevent PvRBP2b from binding TfR1-Tf by causing a collision with the reticulocyte membrane.

279280 enhances PvRBP2b binding to TfR1-Tf in both the reticulocyte binding assay and FRET assay. By HDX-MS, 279280 binds PvRBP2b in a region directly opposite the TfR1 binding site and it could stabilize this region to make it favorable for TfR1 binding. These epitopes that elicit enhancing antibodies are important to elucidate as they would be detrimental to a vaccine antigen.

Combining human mAbs that have different inhibitory mechanisms may have either synergistic or additive effects in *P. vivax* invasion assays. Where inhibitory antibodies compete with each other, choosing the most promising antibody may depend on the antibodies with the fastest association rate. Based on a study of human mAbs targeting the *P. falciparum* invasion ligand PfRh5, fast-associating antibodies were the most efficacious as they could rapidly bind proteins on the merozoite surface before the merozoite invaded the erythrocytes[20]. Based on our results, we propose a combination of four sets of PvRBP2b human mAbs that bind non-overlapping target epitopes, have different structural modes of inhibition, and the fastest association rates. The first set of antibodies includes 241242 or 273264 which binds the N-terminal domain (S168–E460), resulting in steric hindrance with TfR1 and shows association rates of ~3.26 ×10$^5$ (Figs. 2d, 3a, 5b and Supplementary Fig. S2). The second set of antibodies includes 239229 or 262231 that bind in the C-terminal domain (L461-I633), resulting in steric hindrance with TfR1 and association rates of 3.80 × 10$^5$ (Figs. 2d, 3a, 5b and Supplementary Fig. S2). The third combination includes 326327, which causes steric hindrance with the reticulocyte membrane and has an association rate of 3.78 × 10$^5$ (Fig. 2d and Supplementary Fig. S2). The fourth combination antibody is 237235, which is proposed to allosterically affect PvRBP2b binding to Tf and has an association rate of 3.57 × 10$^5$ (Fig. 4 and Supplementary Fig. S2). Although 237235 only inhibits PvRBP2b binding to reticulocytes by 47% (Fig. 2a), it may function similarly to the PfRh5 non-neutralizing antibody R5.011 that acts synergistically with PfRh5 neutralizing antibodies[20]. R5.011 has been shown to slow merozoite invasion and create an opportunity for neutralizing antibodies to bind, although its mechanism of slowing invasion is not known[20]. While PvRBP2b and PfRh5 bind different receptors (PfRh5 binds to basigin), they share a similar structural fold[11,13]. The crystal structure of R5.011 in complex with PfRh5 and a neutralizing antibody, R5.016, shows that R5.011 contacts the N-terminal β-strand and loop at Y155-L162[20,21], and similarly, 237235 forms contacts with the N-terminal loop of PvRBP2b at D173–N174 and Q191–P194 (Fig. 5a). This may suggest that antibodies contacting the N-terminal loop of PfRh5 and PvRBP2b can modulate the binding of these ligands to their receptors. In summary, 241242/273264, 239229/262231, 326327, and 237235 could be examined together in *P. vivax* ex vivo invasion assays to ascertain whether a combination of antibodies is more effective than using single human mAbs for inhibiting parasite invasion.

One of the major challenges in targeting malaria proteins for vaccination is the presence of naturally occurring polymorphisms that may limit the effective use of a vaccine (reviewed in ref. [21]). Our human mAbs were characterized using PvRBP2b sequences from the reference *P. vivax* strain *Salvador I* isolated from El Salvador and it is, therefore, important to investigate the action of these antibodies in Southeast Asian strains of *P. vivax* where our antibodies originate and where polymorphisms may affect antibody function. In particular, the N-terminal domain of PvRBP2b encompassing residues S168–E460 is under balancing selection[11,12]. Targeting multiple PvRBP2b epitopes using a combination of human mAbs may lessen the impact of field polymorphisms that may disrupt antibody binding. Of the four

sets of antibodies that may be useful in combination, structural analyses reveal two field polymorphisms that may affect 241242/273264 binding. 241242/273264 binds in the N-terminal domain where seven polymorphic residues occur and the crystal structures reveal their binding may be affected by K363 and G382 polymorphisms (Supplementary Fig. S9). As the precise epitopes of 239229/262231 and 326327 are unknown, we cannot predict how polymorphisms may affect their binding. Further mutational studies will be needed to confirm whether these human mAbs are affected by existing field polymorphisms. Of the four antibody sets, only 237235 binds a highly conserved region within the N-terminal domain. The only polymorphism within close proximity is R242, which is located directly adjacent to the 237235 epitope (Supplementary Fig. S9). In future studies, it would be important to isolate and examine more human mAbs that bind within the C-terminal region, which has only four polymorphisms and in particular no polymorphisms that reside in the TfR1 interaction site[11,12].

Our structural and functional characterization of PvRBP2b human mAbs provides us with an improved understanding of how the human immune system recognizes PvRBP2b during natural infection with *P. vivax* and the presence of functional blocking antibodies. Furthermore, our interaction assays enable us to distinguish antibodies with different modes of inhibition and prioritize those that may be used in combination to block parasite invasion.

## Methods

**Study population.** Blood samples were obtained from Cambodians with documented *P. vivax* malaria residing in Pursat Province[16]. Large volume blood draws were previously obtained to isolate peripheral blood mononuclear cells (PBMCs) using standard procedures. Plasma samples from individuals with cryopreserved PBMCs were screened for blocking Abs to PvRBP2b. Institutional review boards from the United States National Institutes of Health (NIAID protocol #08-N094, Clinicaltrials.gov NCT00663546), Cambodian Ministry of Health, and University Hospitals of Cleveland Medical Center approved the protocols (IRB No. 04-14-19) for blood collections. Study protocols were approved by the National Human Research Ethics Committee of the Ministry of Health of Brazil (approval No.551/2010). Written informed consent was obtained from all study participants or their parents/guardians.

**Memory B-cell sorting.** To generate tetramers, premium-grade allophycocyanin-labeled streptavidin (SA-APC) (Invitrogen Molecular Probes) was mixed on ice with mono-biotinylated PvRBP2b$_{161-1454}$ with a BirA site using a PvRBP2b$_{161-1454}$: SA-APC ratio of 4:1[22], modified with five additions of SA-APC at 20 min intervals and mixing at 10 min intervals. Aggregates were removed with 17,000 × *g* centrifugation for 20 min at 4 °C. All tetramers were prepared fresh for each experiment. Single cells were identified and sorted following similar previously described techniques, ex vivo from cryopreserved PBMC without activation[16]. Cryopreserved PBMC aliquots were thawed, placed in 10 mL 37 °C RPMI1640 + 10% FBS containing DNase type 1, centrifuged at 400 × *g*, and resuspended in PBS pH 7.2, 0.5% BSA, and 2 mM EDTA. B cells were enriched using immunomagnetic positive selection with anti-CD19 magnetic MACS beads (Miltenyi Biotec). Cells were washed twice with 5 mL FACS buffer with 3 mM EDTA and adjusted to a cell density of 1–2 × 10$^6$ cells/mL. The gating strategy used for sorting PvRBP2b-specific memory B cells is summarized in Supplementary Fig. S1. Cells were stained with mouse anti-human CD20 (PE-Cy5.5, Invitrogen) and anti-human IgG Abs (PE-Cy7 clone G18-145, BD Biosciences) along with SA-APC PvRBP2b$_{161-1454}$ tetramers (Invitrogen, Molecular Probes) and SYTOX Green Dead Cell Stain (Invitrogen) to gate out dead cells. Stained CD19$^+$ cells were filtered through 35 μm nylon mesh and sorted on a BD FACSAria II equipped with chilled stage sorting based on size and complexity. Doublet discrimination was performed to exclude aggregated cells. Individual PvRBP2b-specific CD20$^+$, IgG$^+$ memory B cells were single-cell sorted using a 100 μm nozzle at 20 psi with the following parameters: yield mask 0/32, purity mask 32/32, and phase mask 0/32. Individual cells were sorted directly into 4 μL mRNA extraction buffer on a cooled 96-well metal block. Plates were frozen immediately on dry ice and stored at −80 °C until further processing.

**cDNA synthesis and Ig gene amplification.** The 96-well plates with single cells were thawed on ice; a total cold volume of 7 μL containing 300 ng random hexamers (Qiagen Operon), 12 U RNasin (Promega), and 0.9% NP-40 (Thermo Scientific Pierce) was added to each well. After thorough pipetting and rinsing, wells were capped, centrifuged at 4 °C, heated to 68 °C in a thermal cycler for 5 min, and placed on ice for at least 1 min. Reverse transcription was performed with the addition of 7 μL containing 3.6 μL 5× reverse transcriptase buffer, 10 U RNAsin (Promega), 62 U Superscript III RT (Invitrogen), 0.62 μL dNTPs 25 mM each (Omega Bio-Tek) and 1.25 μL 0.1 M DTT (Sigma). All wells were capped, the plate vortexed in a cold rack for 10 s before centrifugation at 300 × *g*. Thermal cycler conditions for reverse transcription were as follows: 42 °C 5 min, 25 °C 10 min, 50 °C 60 min, 94 °C 5 min and 4 °C hold. When completed, 10 μL of nuclease-free PCR water was added to each well. Immediately following cDNA synthesis, IgG genes (Igg) were amplified in a total volume of 20 μL per well for the first round of nested PCR, for IgG heavy chain (Iggh), IgG kappa (Iggκ) and IgG lambda (Iggλ), using previously described primers[23]. In brief, a master mix was prepared consisting of 15.58 μL water, 2 μL 10× HotStar PCR buffer (Qiagen), 0.065 μL 5′ primer mix, 0.065 μL 3′ primer, 0.2 μL dNTP solution, and 0.09 μL HotStarTaq per well, to which 2 μL cDNA from individual sorted B cells were added and Igg amplified under the following conditions: thermal cycle PCR at 94 °C for 15 min; 50 cycles at 94 °C for 30 s, 58 °C (Iggh and Iggκ) or 60 °C (Iggλ) for 30 s, then 72 °C for 55 s; one cycle at 72 °C for 10 min. Second round of nested PCR for Iggh, Iggκ, and Iggλ used 2 μL of first-round PCR product with second-round primers and the same master mix protocol, with the following conditions: thermal cycle PCR at 94 °C for 15 min; 50 cycles at 94 °C for 30 s, 58 °C (Iggh and Iggκ) or 60 °C (Iggλ) for 30 s, 72 °C for 45 s; one cycle at 72 °C for 10 min. The PCR product generated was purified and sequenced, with V(D)J genes determined using IMGT/V-Quest. Primers with specific restriction enzyme sites for V and J regions were used to amplify the first-round PCR product to generate a fragment for cloning[16]. PCR product was purified, restriction enzyme digested, cloned into Iggh, Iggκ or Iggλ expression vectors, and chemically transformed into 5 μL aliquots of TOP10 *E. coli* cells (ThermoFisher Scientific). Successful transformants were screened by PCR amplification utilizing a vector-specific primer paired with an insert-specific primer, sequenced and compared to the second-round PCR product sequence. Primers used are listed in Supplementary Table S1.

**Definition of clonal groups.** Clonal groups were based on heavy-chain nucleotide sequences. Any PCR product with >0.8% nucleotide sequences with a Phred score <20 was excluded. From PCR-amplified sequences, we determined heavy-chain alleles using IMGT/V-QUEST (http://www.imgt.org, international ImMunoGeneTic information system). Due to primer mixture ambiguities, the first 20–22 nucleotides of IgGH variable regions (IgGHV) were designated as germline, thus this region was not evaluated for somatic hypermutations. IMGT/V-QUEST was used to assign V(D)J organization, and sequences were grouped by shared IgGHV genes and CDR3 length. The clonal grouping was determined using Sequence Manipulation Suite: Ident and Sim using Ab-specific clusters as previously defined[16]. A clonal group is defined by the same IgGHV gene and CDR3 length, along with >72% similarity of each CDR3 amino acid sequence.

**Expression and purification of recombinant proteins.** The constructs were derived from the *P. vivax* strain *Salvador I*, obtained from PlasmoDB Database (www.plasmodb.org; accession number: PVX_094255, total length 2806 amino acids). PvRBP2b$_{161-470}$ was expressed using *E. coli* strain SHuffle® T7 (New England Biolabs) and Terrific Broth (TB) supplemented with 100 μg/mL of carbenicillin. Flasks containing 1 L of medium were incubated in Multitron shaker (Infors HT) at 37 °C at 180 rpm. At OD 600 of around 1.0, IPTG (Astral) was added to a final concentration of 1.0 mM and protein expression was allowed to continue for 20 h at 16 °C. Cells were harvested by centrifugation at 7000 × *g*, resuspended in freezing buffer containing 50 mM Tris-HCl pH 7.5, 500 mM NaCl, 10% (v/v) glycerol supplemented with cOmplete EDTA-free protease inhibitor cocktail (Roche), flash-frozen in liquid nitrogen and stored at −80 °C until further processing. For the purification, the cell pellet was thawed on ice and resuspended in the freezing buffer supplemented with 0.5 mg/mL of DNase and 1.0 mg/mL of lysozyme (both from Sigma-Aldrich). Cells were lysed using sonicator Sonopuls UW 3200 (Bandelin) equipped with VS 70 T probe. The resulting crude cell extract was clarified by centrifugation at 30,000 × *g* for 45 min at 4 °C. The supernatant was applied on a 5 mL HisTrap column (GE Healthcare) pre-equilibrated with the freezing buffer then washed using at least 10 column volumes of the washing buffer: 50 mM Tris-HCl pH 7.5, 500 mM NaCl, 10% (v/v) glycerol, and 10 mM imidazole. The bound protein was eluted from the column using the same buffer but containing 300 mM imidazole. The eluted fractions were pooled and dialyzed overnight in the presence of TEV protease into 20 mM HEPES pH 7.0 and 100 mM NaCl. The resulting protein sample was applied on a 5 mL Q-Sepharose HiTrap column (GE Healthcare) pre-equilibrated with the dialysis buffer. Unbound material was washed away using at least 10 column volumes of the buffer. The protein was eluted from the column using a gradient of 20 mM HEPES pH 7.0 and 1.0 M NaCl. Collected fractions were analyzed on SDS-PAGE and those containing the protein of interest were concentrated using an Amicon Ultra-4 10 kDa molecular weight cut-off concentrator (Millipore) and injected onto S75 Superdex 16/600 size exclusion column (GE Healthcare) pre-equilibrated with 20 mM HEPES pH 7.5 and 150 mM NaCl. The monodisperse peak fractions containing protein were pooled and concentrated using the same type of concentrator and used for crystallization trials. Expression and purification of PvRBP2b$_{161-1454}$, PvRBP2b$_{474-1454}$, PvRBP2b$_{161-969}$, PvRBP2b$_{169-813}$, PvRBP2b$_{169-652}$, and PvRBP2b$_{169-470}$ were performed in a similar manner as described above or previously[2,5,24].

A construct encoding the soluble ectodomain of human transferrin receptor 1 (TfR1, residues 121–760) was obtained from Addgene (pAcGP67A-TfR, Plasmid #12130[25]). The TfR1 sequence follows the leader peptide from the baculovirus protein GP67, a 6xHis-tag, and a factor Xa cleavage site in a modified form of the pAcGP67A expression vector (Pharmingen) as described before[25]. TfR1 was expressed in a lytic baculovirus/insect cell expression system using Sf21 cells (Life Technologies) and Insect-XPRESS™ Protein-free Insect Cell Medium supplemented with L-glutamine (Lonza). Protein expression was induced by inoculation of the cell culture at around $1 \times 10^6$ cells/mL with the third passage stock (P3) of the virus and allowed to progress for three days. The supernatant was separated from the cells by centrifugation at $2000 \times g$ for 20 min and concentrated using a tangential flow filtration device equipped in a cassette with 10 kDa molecular weight cut-off (Millipore). Concentrated supernatants were subsequently dialyzed into a buffer containing 20 mM Tris-HCl pH 7.5 and 300 mM NaCl in order to remove the residuals of the culture media that might interfere with the subsequent steps of purification. The supernatant was applied on a 5 mL HisTrap column (GE Healthcare) pre-equilibrated with the dialysis buffer then washed using at least 10 column volumes of the dialysis buffer supplemented with 20 mM imidazole. The bound protein was eluted using the same buffer but containing 300 mM imidazole, then concentrated using an Amicon Ultra-4 30 kDa molecular weight cut-off concentrator (Millipore) and injected onto S200 Superdex 16/600 size exclusion column (GE Healthcare) pre-equilibrated with 20 mM HEPES pH 7.5 and 100 mM NaCl and 50 mM NaHCO3. The final samples were supplemented with 10% (v/v) glycerol, flash-frozen in liquid nitrogen and stored at −80 °C.

Apo-transferrin purified from human serum was purchased from Sigma-Aldrich (Catalog Number T4382). The protein powder was resuspended in a buffer containing 100 mM disodium carbonate at pH 5.9. Loading of transferrin with iron was performed as described in the manufacturer's leaflet by adding a solution of ferrous ammonium sulfate, hexahydrate (Sigma-Aldrich) corresponding to 2% (w/w) of protein mass. The sample was incubated with stirring for 1 h at room temperature. The pH was then raised to 8.5 using 1.0 M disodium carbonate and the solution was mixed for an additional 2 h. Holo-transferrin was subsequently re-purified by gel filtration chromatography using an S200 Superdex 16/600 size exclusion column (GE Healthcare) equilibrated with 20 mM NaHEPES pH 7.5, 100 mM NaCl and 50 mM NaHCO3 buffer. The final samples were supplemented with 10% (v/v) glycerol, flash-frozen in liquid nitrogen, and stored at −80 °C.

**Antibody expression and purification.** Recombinant human mAbs were expressed in Expi293 HEK cells (Life Technologies), which were maintained in suspension at 37 °C and 8% CO2. Cells were transfected at a density of $3 \times 10^6$ with equal amounts of heavy and light-chain paired plasmids using polyethyleneimine (PEI, Sigma-Aldrich) at a ratio of 1:03 of the total amount of plasmid to PEI. One day after transfection, valproic acid was added to cultures to a final concentration of 0.025 M. Seven days after transfection, the supernatant was collected by centrifugation and filtered through a 0.22 μm filter. Human mAbs were purified by loading the supernatant onto a 1 mL Protein A HP HiTrap column (GE Healthcare). Columns were equilibrated and washed using Dulbecco's phosphate-buffered saline (DPBS). Human mAbs were eluted using 0.1 mM citric acid pH 3.00 and neutralized with 1 M Tris-HCl pH 9.0. A second purification step was performed by loading Protein A eluate on a Hiload 16/600 Superdex 200 pg gel filtration column (GE Healthcare), which was equilibrated and run using DPBS. Human mAbs were concentrated using Amicon Ultra-04 5 kDa (Millipore). Antibody concentration was determined by absorbance measurement at 280 nm using a Nanodrop and purity was determined using SDS-PAGE. Human mAbs Fab fragment purification and complex formation with PvRBP2b169–470 were performed as described previously[2].

**Antibody specificity ELISA.** 96-well flat-bottomed plates were coated with 65 nM of recombinant protein in 100 μL of PBS at 4 °C overnight. All washes were done three times using PBS and 0.1% Tween and all incubations were performed for 1 h at room temperature. Coated plates were washed and blocked by incubation with 10% skim milk solution. Plates were washed then human mAbs at 1 μg/mL were added and incubated, then washed and incubated with horseradish peroxidase (HRP)-conjugated goat anti-human secondary antibody (1:500, Jackson ImmunoResearch). After a final wash, 100 μL of azino-bis-3-ethylbenthiazoline-6-sulfonic acid (ABTS liquid substrate; Sigma) was added and incubated in the dark at room temperature and 100 μL of 1% SDS was used to stop the reaction. Absorbance was read at 405 nm and all samples were done in duplicate.

**Bio-layer interferometry.** Antibody affinities were measured using an Octet RED96 instrument. Assays were performed at 25 °C in solid black 96-well plates agitated at 1000 r.p.m. The kinetic buffer was composed of PBS 0.1% BSA, 0.05% TWEEN. A 60 s biosensor baseline step was applied before human mAbs were loaded onto anti-human IgG Fc capture sensor tips (AHC) by submerging sensor tips in 5 μg/mL human mAb until a response of 0.5 nm then washed in a kinetic buffer for 60 s. Association measurements were performed using a two-fold concentration gradient of PvRBP2b161–1454 from 6.25–200 nM for 200 s and dissociation was measured in a kinetic buffer for 300 s. Sensor tips were regenerated using a cycle of 5 s in 10 mM glycine pH 1.5 and 5 s in kinetic buffer repeated five times. Baseline drift was corrected by subtracting the average shift of a human mAb

loaded sensor not incubated with PvRBP2b161–1454, and an unloaded sensor incubated with PvRBP2b161–1454. Curve fitting analysis was performed with Octet Data Analysis 10.0 software using a global fit 1:1 model to determine $K_D$ values and kinetic parameters. Curves that could not be reliably fitted were excluded from further analysis.

**Avidity ELISA.** ELISA was carried out to estimate the avidity and overall stability of antigen–antibody complexes using the chaotropic agent ammonium thiocyanate (NH4SCN) (Acros Organics Fisher Scientific). The plates were coated with 50 ng of PvRBP2b161–1454 in PBS and incubated overnight. After washing and blocking the plates, the wells were incubated with 50 μL/well of human mAbs at 0.2 μg/mL for 1 h at room temperature. After three washings with PBST (0.05% Tween 20), plates were treated with PBS, 0.5 M, 1 M, and 2 M of NH4SCN for 20 mins at room temperature. Following washing three times with PBST, wells were incubated with HRP labeled anti-human IgG (Fc) antibody (1:1000 dilution, Jackson ImmunoResearch) in PBS supplemented with 1% (w/v) bovine serum albumin (BSA) for 1 h at room temperature. Plates were developed using 100 μL/well of 3,3′,5,5′-tetramethylbenzidine (TMB, ThermoFischer Scientific) substrate for 20 min at room temperature. The reaction was stopped by adding 50 μL/well of 10% sulfuric acid and the optical densities were read at 450 nm. The results were expressed as avidity index (AI). The percentage of AI was calculated as follows: OD of wells treated with NH4SCN divided by OD of wells without treatment multiplied by 100. All measurements were performed in duplicate.

**Flow cytometry-based reticulocyte-binding assay.** The cord blood for reticulocyte binding assays was obtained through an MTA (ID# M19/110) with the Bone Marrow Donor Institute (BMDI) Cord Blood Bank. The human ethics project "14/09, Malaria parasite growth and invasion into reticulocytes" was approved by the Walter and Eliza Hall Institute Human Research Ethics Committee. Reticulocytes enriched from cord blood were resuspended in PBS to a final volume of $0.5 \times 10^7$ cells/mL. PvRBP2b161–1454 at 4 μg/mL was incubated together with human mAbs at a 5-fold molar excess in 50 μL of the resuspended reticulocytes for 1 h at room temperature. All washes were performed with PBS supplemented with 1% (w/v) BSA and centrifuged at $2000 \times g$ for 1 min. All incubations were for 1 h at room temperature unless otherwise specified. Samples were washed once and incubated with PvRBP2b rabbit polyclonal antibodies, 100 μL at 12.5 μg/mL, then washed and incubated with Alexa Fluor 647 chicken anti-rabbit secondary antibody (1:100; Life Technologies). After one final wash, 100 μL thiazole orange (TO) (Retic-Count Reagent; BD Biosciences) was added and incubated for half an hour. Samples were resuspended in 200 μL PBS and data collected on the LSR II flow cytometer (BD Biosciences). 50,000 events were recorded and results were analyzed using FlowJo software v.10 (Three Star). The gating strategy used is shown in Supplementary Fig. S3. The background signal from a control sample with rabbit polyclonal antibody and Alexa Fluor 647 conjugated antibody and without protein was subtracted from all binding results. To enable comparisons between biological replicates, the percentage of PvRBP2b161–1454-bound cells in the presence of human mAbs was divided by the percentage of PvRBP2b161–1454-bound cells with no human mAbs and multiplied by 100 to obtain the percentage of PvRBP2b161–1454-bound cells relative to the no human mAbs control.

**Fluorescence resonance energy transfer (FRET) assay.** PvRBP2b161–1454 and TfR1 were labeled with N-hydroxysuccinimide ester-activated DyLight 488 (DL488) and DyLight 594 (DL594) (Life Technologies) respectively. The dyes were dissolved in dimethyl sulfoxide (DMSO) (Sigma) and added at five-fold molar excess to the protein being labeled. After 1-h incubation at room temperature, the un-conjugated dye was removed using a Micro Bio-Spin P-6 column (BioRad) at $1000 \times g$, 2 min. The average dye per protein was ~3.5 dye/protein for PvRBP2b161–1454 and ~1.8 dye/protein for TfR1. PvRBP2b161–1454-DL488, TfR1-DL594, and Tf were mixed in a 1:1:1 molar ratio with a 5-fold excess of human mAbs in a final reaction volume of 10 μL in FRET buffer (50 mM Tris-HCl pH 6.8, 150 mM NaCl, 0.01% BSA) read in Corning 384-well plates and each sample was performed in triplicate. 1 μL of 2% SDS was added to one triplicate sample with no human mAbs to measure the background signal. Fluorescence intensity was measured using EnVision plate reader (PerkinElmer Life Sciences). DL488 (donor) fluorescence was measured with a 485/14-nm excitation filter and 535/25-nm emission filter and DL594 (acceptor) fluorescence was measured with a 590/20-nm excitation filter and 615/9-nm emission filter. Sensitized emission was measured with a 485/14-nm excitation filter and 615/9-nm emission filter. Fluorescence measures were analyzed using Prism software (GraphPad; version 8.4.3). To account for bleed-through of dyes into the sensitized emission spectra, standard curves for both dyes were measured and the raw data were transformed by multiplying with the slope of the standards. The y-intercept from the standard curves indicated the background fluorescence when no dye was present. The FRET signal was calculated with the following equation: FRET signal = Raw sensitized emission – transformed DL488 emission – transformed DL594 emission – DL488 standard curve y-intercept – DL594 standard curve y-intercept. To compare between biological replicates, the percentage FRET signal relative to the no inhibitor control was calculated by dividing the FRET signal with inhibitor by the FRET signal with no inhibitor, multiplied by 100.

**Immunoprecipitation assays**. Immunoprecipitation assays were performed using equal amounts of PvRBP2b$_{161-1454}$, TfR1, Tf, and human mAbs in a 100 μL reaction volume in PBS. After incubation for 1 h at 4 °C on rollers, 15 μL of packed Protein G-sepharose beads was added to bind human mAbs and incubated overnight at 4 °C on rollers. Beads were washed three times in PBS by centrifugation at $2000 \times g$ for 1 min. Human mAbs and associated proteins were eluted from the beads by boiling for 5 min in 2× SDS-PAGE reducing sample buffer and analyzed on 4–12% Bis-Tris gels (Life Technologies) run in MOPS buffer at 180 V for 60 min. Protein bands were visualized by staining with SimplyBlue SafeStain (Life Technologies).

**Competition ELISA**. For the competition ELISA, the method above was performed with the following modifications. 96-well flat-bottomed plates were coated with 100 μL of antibody at 1.5 ng/μL overnight. Competing antibodies were pre-incubated with PvRBP2b$_{161-1454}$ at a 20:1 molar excess, then incubated with antibody-coated plates. For competition between human mAbs, the detection was performed using HRP-conjugated streptavidin binding to biotinylated PvRBP2b$_{161-1454}$. For competition between human mAbs and mouse mAbs, the detection was performed using HRP-conjugated goat anti-human secondary antibody (1:4000). Detection was performed as described in the ELISA method above.

**Hydrogen–deuterium exchange coupled to mass spectrometry (HDX-MS)**. HDX-MS experiments were performed at the UniGe Protein Platform (University of Geneva, Switzerland). A similar protocol described previously was applied[26]. HDX reactions were done in 50 μL volumes using 50 pmol of PvRBP2b$_{161-1454}$ and a 1.3 fold molar excess of a human antibody. Reaction with every antibody was performed following exactly the same procedure. Briefly, RBP2b was pre-incubated with the antibody on ice for 30–60 min. 7 μL of the RBP2b-mAb mix was pipetted into a tube and left to equilibrate at the experimental temperature for 10 min. Deuterium exchange reactions were initiated by adding 43 μl of D$_2$O exchange buffer (82.5% D$_2$O, 10 mM Hepes pH 7.5, 150 mM NaCl) to the 7 μL RBP2b-mAb mixture. Reactions were carried out for 5 min, either at 0 °C or at 22 °C and terminated by the sequential addition of 20 μL of ice-cold quench buffer 1 (6 M Urea/ 0.1 M NaH$_2$PO$_4$ pH 2.5/1% Formic Acid). Samples were immediately frozen in liquid nitrogen and stored at −80 °C for up to 4 weeks. All experiments were repeated in triplicates.

To quantify deuterium uptake into the protein, protein samples were thawed and injected in the UPLC system immersed in ice. The protein was digested via two immobilized pepsin columns (Thermo #23131), and peptides were collected onto a VanGuard precolumn trap (Waters). The trap was subsequently eluted and peptides separated with a C18, 300 Å, 1.7 μm particle size Fortis Bio column 100 × 2.1 mm over a gradient of 8–30% buffer B over 20 min at 150 μL/min (Buffer A: 0.1% formic acid; buffer B: 100% acetonitrile). Mass spectra were acquired on an Orbitrap Velos Pro (Thermo), for ions from 400 to 2200 $m/z$ using an electrospray ionization source operated at 300 °C, 5 kV of ion spray voltage. 282 peptides were identified by data-dependent acquisition after MS/MS and data were analyzed by Mascot. An 85% sequence coverage was obtained. Deuterium incorporation levels were quantified using HD examiner software (Sierra Analytics), and the quality of every peptide was checked manually. Results are presented as a percentage of maximal deuteration using a highly deuterated sample prepared by incubating the protein for 1 h in 1 M Gdn-HCl before incubation for 6 h in a deuterated buffer. The average standard deviation for percentage deuteration was 3.4%. Changes in deuteration level between two states were considered significant if >12% and >0.6 Da and $p < 0.05$ (unpaired $t$-test). Raw data for percentage deuterium incorporation for all peptides can be found in Supplementary Data 2 and 3. Differences in a number of deuterons incorporated and differences in percentage deuteration for PvRBP2b peptides alone and in complex with PvRBP2b human mAbs for all peptides is shown in Supplementary Data 4 and 5.

**Protein crystallization**. Large-scale, sparse-matrix sitting-drop crystallization trials were performed at the CSIRO Collaborative Crystallization Center (CSIRO; Parkville, Australia). Crystals of PvRBP2b-237235, PvRBP2b-243244, PvRBP2b-251249 and PvRBP2b-273264 were harvested directly from the sitting-drop plates, while PvRBP2b-241242, PvRBP2b-253245, PvRBP2b-258259, and PvRBP2b-283284 crystals were further refined in-house by hanging-drop vapor diffusion crystallization trials. All crystals were obtained at 295 K. Crystals of 237235 in complex with PvRBP2b$_{169-470}$ were obtained in 0.1 M magnesium formate, 15% (w/v) PEG 3,350 at 10 mg/mL of complex. Crystals of 241242 in complex with PvRBP2b$_{169-470}$ were obtained in 400 mM sodium thiocyanate, 17% (w/v) PEG 3350 at 5.5 mg/mL of complex. Crystals of 243244 in complex with PvRBP2b$_{169-470}$ were obtained in 0.2 M trimethylamine N-oxide, 0.1 M Tris chloride pH 8.5, 20% (w/v) PEG monomethyl ether 2000 at 13 mg/mL of complex. Crystals of 251249 in complex with PvRBP2b$_{169-470}$ were obtained in 0.2 M DL-malate-imidazole pH 6.0, 15% (w/v) PEG 4000 at 3 mg/mL of complex. Crystals of 253245 in complex with PvRBP2b$_{169-470}$ were obtained in 0.2 M sodium dihydrogen phosphate pH 4.0, 16% (w/v) PEG 3350 at 5 mg/mL of complex. Crystals of 258259 in complex with PvRBP2b$_{169-470}$ were obtained in 0.4 M ammonium nitrate, 18% (w/v) PEG 3350 with crystal seeds from the same condition at 7 mg/mL of complex. Crystals of 273264 in complex with PvRBP2b$_{169-470}$ were obtained in 0.1 M sodium cacodylate pH 6.0, 0.1 M magnesium chloride, 15 % (w/v) PEG 4,000 at 8 mg/mL of complex. Crystals of 283284 in complex with PvRBP2b$_{169-470}$ were obtained in 0.3 M ammonium nitrate, 0.1 M sodium HEPES pH 7.0, 26% (w/v) PEG 2000 at 11 mg/mL of complex.

**Data collection and structure determination**. Crystals were cryo-protected in mother liquor containing 20% (v/v) glycerol and flash-frozen in liquid nitrogen. Diffraction data were collected at the MX2 beamline at the Australian Synchrotron (Clayton, Australia) at 100 K ($\lambda = 0.9537$ Å) using an Eiger X 16 M pixel detector[27]. Diffraction data were processed with the XDS package[28] (Version Jan 31, 2020 BUILT = 20200131). The highest resolution shell included for processing was determined based on data completeness of >95% and CC$_{1/2}$ of >50%. The probable number of molecules in the asymmetric unit was calculated using the program Matthews[29]. Molecular replacement was performed with Phaser (CCP4 suite; version 7.0.052)[30,31] using the structure of PvRBP2b$_{169-470}$ (PDB accession code 5W53) and unrelated Fab antibody fragments as search models. To obtain the best starting search model for antibody Fabs, the online tool BLASTp (https://blast.ncbi.nlm.nih.gov/) was used to search the Protein Data Bank (PDB) for antibody structures with the highest sequence similarity to the PvRBP2b human mAbs and the best structural refinement statistics. Heavy and light-chain antibody sequences were queried separately then combined into one Fab to use as a search model. The models used were: 237235/243244—4HWE for both heavy and light chain; 241242—5ODB for both heavy and light chain, 251249; 6B0S for the heavy chain and 5I16 for the light chain; 253245—6RCO for the heavy chain and 4UNU for the light chain; 258259/273264—6AZM for the heavy chain and 5IFH for light chain, and 283284—6MLK for the heavy chain and 5E08 for the light chain. The angle between constant and variable regions of PvRBP2b antibody Fabs was different from the model, so to obtain a better molecular replacement solution, antibody constant regions were deleted from the first molecular replacement solution, and Phaser was run again using the antibody variable region bound to PvRBP2b$_{169-470}$ as one search model and an antibody constant region as a subsequent search model. In the subsequent Phaser run, for antibodies with lambda light chains (237235, 241242, 243244, 253245, 258259, 273264) the antibody constant region with the PDB accession code 4LLD was used as a search model, and for antibodies with kappa light chains (251249 and 283284), the antibody constant region in PDB accession code 5NYX was used as a search model. Structures were manually improved with Coot[32] and refined using Phenix[33] (version 1.14) in an iterative process. To improve the refinement, the resolution cut-off for diffraction data for 253245 was extended to 3.48 Å to make more use of weak data in the outer shell (3.69–3.48 Å)—within this outer resolution shell, CC$_{1/2}$[34] remained significant at the $P = 0.001$ level and data completeness remained above >90%. Pseudo-merohedral twinning was detected in the 253245 data set by phenix.xtriage. 253245 was therefore refined in the presence of the twin operator (h, -k, -h-l), yielding an associated twin fraction of 0.15. Crystallographic data collection and refinement statistics for all structures are summarized in Supplementary Table S2. Interactions between PvRBP2b and antibody Fabs was determined using PISA[35] (CCP4 suite; version 2.1.0) and summarized in Supplementary Table S3. Figures were prepared using Pymol (http://www.pymol.org).

**Statistical analysis**. Prism (version 8.4.3) was used to perform one-way ANOVA followed by Dunnett's multiple comparisons test using both 043038 and 099100 as controls.

**Reporting summary**. Further information on research design is available in the Nature Research Reporting Summary linked to this article.

## Data availability

*P. vivax* reticulocyte binding protein 2b constructs were derived from the *P. vivax* strain *Salvador I* from the PlasmoDB Database (www.plasmodb.org; accession number: PVX_094255). The structure models used in molecular replacement are from the Protein Data Bank under the following accession codes: 4HWE, 5ODB, 6B0S, 5I16, 6RCO, 4UNU, 6AZM, 5IFH, 6MLK, 5E08, 4LLD, and 5NYX. Coordinates and structure factors have been deposited in the Protein Data Bank under accession codes 6WM9 (PvRBP2b-237235 complex), 6WN1 (PvRBP2b-241242 complex), 6WNO (PvRBP2b-243244 complex), 6WOZ (PvRBP2b-251249 complex), 6WTY (PvRBP2b-253245 complex), 6WTV (PvRBP2b-258259 complex), 6WTU (PvRBP2b-273264 complex), and 6WQO (PvRBP2b-283284 complex). Source data are provided with this paper.

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

## Acknowledgements

We thank Sokunthea Sreng, Seila Suon, Chanaki Amaratunga, and Rick M. Fairhurst who identified *P. vivax* immune subjects acquired necessary approval and helped the PBMC collection. We thank Janet Newman and Bevan Marshall from the CSIRO Collaborative Crystallization Centre (www.csiro.au/C3) (CSIRO; Parkville, Australia) for assistance with setting up the crystallization screens. We thank Mike Lawrence for his invaluable advice and assistance in the improvement of the structure refinement for antibody complex PvRBP2b-253245. This research was undertaken in part using the MX2 beamline at the Australian Synchrotron, part of ANSTO, and made use of the Australian Cancer Research Foundation (ACRF) detector. We also thank the MX2 beamline staff at the Australian Synchrotron for their assistance during data collection. We thank Alexandre Hainard, from the Proteomics Platforms at the University of Geneva, for assistance with HDX-MS data acquisition. O.V. was funded by Carigest SA (to D.S.-F.) R.V. was funded by the European Research Council under the European Union's Horizon 2020 Research and Innovation program under grant agreement no. 695596 (to D.S.-F.). W.-H.T. is a Howard Hughes Medical Institute-Wellcome Trust International Research Scholar (208693/Z/17/Z) and supported by the National Health and Medical Research Council of Australia (GNT1143187, GNT1160042, GNT1160042, GNT1154937).

## Author contributions

A.G., C.T.F., and L.L.C. sorted B cells, analyzed antibody sequences, and cloned antibody variable regions. A.G. measured antibody avidity and performed competition experiments with PvRBP2b human mAbs. L.-J.C. performed the in vitro functional assays, affinity measurements, mapping of PvRBP2b human mAbs, and antibody competition experiments with mouse mAbs. L.-J.C. expressed and purified all PvRBP2b human mAbs and prepared samples for protein crystallography, L.-J.C. and M.H.D. crystallized PvRBP2b-Fab complexes, collected diffraction data, and determined the crystal structures. S.M. assisted in the purification of recombinant PvRBP2b fragments. O.V. and R.V. performed the HDX-MS sample preparation and acquisition data analysis. D.S.-F., I.M., W.-H.T., and C.L.K. helped design the experiments.

## Competing interests

The authors declare no competing interests.
