## [Peer Review File · Nature Communications]

Reviewers' Comments:

Reviewer #1:

Remarks to the Author:

This paper by Chan et al describes naturally occurring antibodies to *P. vivax* and it is an interesting contribution to the field of knowledge and there is a lot of data in the paper. The isolation and characterization of these antibodies will be important for further studies.

However, there are issues that I think needs some clarification. The authors have performed both ammonium thiocyanate assays and bio-layer interferometry for measurement of avidity, but I can't really see any correlation analysis comparing with the rest of the data. So does it make a difference whether the antibodies bind with high or low avidity? And which of the two methods (if any) match the other results? When looking at the data in supplementary fig S2, all the K_a values look very similar but K_d values differ a lot between high and low values. If an antibody binds with a good K_d , it should be able to stay bound to its target for a longer time period and hence be more effective in it's function?

Transferrin is a protein that circulates in plasma. What about binding of the antibodies to free transferrin, compared to in complex with TfR1? If all antibodies are bound to free Transferrin, perhaps none will bind to the complex?

Something that is not discussed in this paper is the presence of iron. The affinity of TfR1 for Transferrin changes in the presence of iron and there is also a pH-dependent cooperativity of lobes of transferrin. This might have a big impact on binding of antibodies. How much iron is present in the different assays? Is the pH the same in all assays?

The issue of iron is my major concern, since the whole system of TfR1 and transferrin is built up for taking care of iron and in a human body in real life it could have a major impact on binding of antibodies.

Reviewer #2:

Remarks to the Author:

In the manuscript "Naturally acquired blocking human monoclonal antibodies to Plasmodium vivax reticulocyte binding protein 2b", Chan et al. characterize a set of naturally acquired human antibodies against a blood-stage antigen of *P. vivax*. The authors use a broad array of biochemical techniques to identify eight PvRBP2b human mAbs that inhibit the binding of PvRBP2b to reticulocytes by blocking complex formation with TfR1 and Tf. A nice trend is established whereas mAbs that compete with each other in ELISA assays share similar phenotypes in the functional assays. The study is further substantiated by extensive molecular insights obtained by HDX-MS and X-ray crystallography. Overall this is an impactful report and comprehensive study that describes the molecular basis of *P. vivax* inhibition by human monoclonal antibodies and suggests possible use of mAbs as therapeutics for *P. vivax* malaria. The manuscript will be of interest to the broad journal readership once the concerns below are addressed:

- Fig. 1A: VH genes should be labeled in the panel, at least for the most frequent ones already highlighted in color.
- The statement "Since antibodies from the same clonal group recognize the same epitope [...]" is unsubstantiated.
- The authors conduct binding affinity and binding avidity measurements, as described in Fig. 1D and 1E. Because the antibodies are not labeled in Fig. 1D, it is impossible to know whether the high-avidity antibodies are also the high-affinity antibodies. It would be helpful if the antibodies could be labeled in Fig. 1D to allow this determination by visual inspection, or perhaps even better if an affinity vs avidity association plot could be added to Fig. 1.
- In Fig. 3B, it is unclear how the dark black boxes presumably delineating different epitope bins were defined. This information should be provided in the figure legend. To this Reviewer, it is unclear that those epitope bin boxes were drawn in the most effective way. For example, couldn't the large top left

box be subdivided into three more precise sub-boxes? Also, a singular box is drawn at the bottom right around mAbs 250233 and 267268. But is there any evidence that these two mAbs actually compete with one another and constitute an epitope bin?

- In the Discussion, the authors propose that three clusters of their inhibitory antibodies may be affected by field polymorphisms found in PvRBP2b. Please highlight the locations of the documented polymorphisms in the figure of the PvRBP2b:Fab complex structures. Additionally, a supplemental figure showing the sequence of PvRBP2b with the polymorphic regions and mAbs binding regions highlighted would be a helpful resource for the reader.

- Another question relating to the naturally occurring polymorphisms: was the PvRBP2b protein construct used in the study typical of a circulating East-Asian *P. vivax* isolate? A lot of questions regarding the versatility of selected antibodies could be answered by performing bio-layer interferometry experiments with PvRBP2b from geographically distant *P. vivax* isolates (from Brazil, etc), if these recombinant proteins are already available.

- In the Discussion, the authors suggest that "241242/273264, 239229/262231, 326327 and 237235 could be examined together in *P. vivax* ex vivo invasion assays to ascertain whether a combination of antibodies is more effective than using single human mAbs for inhibiting parasite invasion." This proposition is attractive; but it is unclear why this experiment wasn't attempted. The impact of the reported biophysical results and overall conclusion of the manuscript would be considerably strengthened should this investigation be completed.

- 279280 binding to PvRBP2b displays signs of antibody-dependent enhancement in FRET assays and flow cytometry-based reticulocyte binding. Can the authors speculate what might be the molecular basis of this phenomenon?

- In the second molecule of the ASU for the PvRBP2b:241242 crystal structure, why is so little of the Fab constant domain for chains E and F built (PDB ID: 6WN1)? The Rwork/Rfree may improve slightly if the remaining model is built into any visible density.

- In Supplementary Table 5 (crystallography table), the authors should report all values to the adequate significant figure, and be consistent across all entries of the same type.

Reviewer #3:

Remarks to the Author:

The manuscript titled "Naturally acquired blocking human monoclonal antibodies to Plasmodium vivax reticulocyte binding protein 2b" is authored by Chan et al. Individuals with higher levels of antibodies to PvRBP2b have been shown to have reduced risk of *P. vivax* infection and disease, however the reason underlying this is still unknown. In this manuscript, the authors report the isolation and characterization of naturally acquired anti-PvRBP2b human monoclonal antibodies (mAbs) from two individuals in an effort to determine and understand the structural mechanisms by which they inhibit binding to PvRBP2. The authors have used an array of biophysical assays, e.g, BLI, FRET, to determine which mAbs bind and their binding affinities to PvRBP2. Coupled with these they have also used X-ray crystallography and solution phase hydrogen-deuterium exchange (HDX), to map the binding sites/epitopes on PvRBP2.

The idea seems fundamentally sound to me, however I am not as familiar with the biology and thus feel most comfortable commenting on the HDX methods and analysis part of the manuscript. Overall I think that the manuscript is well written, the research question is clear and well-addressed. I also think that this study will be of interest to the HDX community because solution-phase HDX is increasingly becoming a tool of choice to probe protein-antibody interactions and map epitopes in order to provide understanding of the underlying basis of such interactions, and is also being implemented in the biopharma industry. I recommend this manuscript for publications after addressing the issues below. Once the concerns below have been properly addressed, this could be a manuscript worthy of publication in Nature Communications.

HDX was performed to map put the epitopes of for PvRBP2-mAbs, by comparing the deuterium uptake data in the presence and absence of the mAbs.

Major issues:

1) HDX methods: I understand that the authors isolated many human mAbs and thus have had to work with many mAbs. So I appreciate the time and effort they put in performing HDX experiments with these many mAbs. However it would benefit the readers if the authors justified their selection of 1 time points (5 mins). In fact choosing more time points and a longer time point than 5 mins, would have benefitted the authors. I am in no way suggesting that the authors redo the experiment, however one big caveat of having one time point is that if a region does change in the bound state, the change might not take place at the time point chosen (i.e. 5 mins) for the experiment. For e.g., if a protein that binds an antibody has 7 regions that change, HDX at one single time point may pick up only 4 out of those 7 regions because the other 3 happen are either at a fast or a slow time scale. So those 3 regions may appear as they have not changed. Thus no change actually means that there is no observed change in HDX, because the change happened outside of the time frame of the experiment.

(a) With this in mind, I suggest that the authors re-write the sentence on Page#10 "Protein amide protons were labelled with deuterium for five min at either 0 °C or 22°C in deuterated buffer, providing information on the fast-exchanging and slow-exchanging regions of the protein." Because there will be way slower exchanging regions which will not begin exchanging in 5 minutes, & may take up to several hours.

(b) On page #11 where the authors write that "Antibody 281282 likely binds very similarly to 251249 and 252248 as all three antibodies have the same profile in the competition assay (Fig. 3B), but no HDX changes were observed in the presence of antibody 281282 (Fig. 4A). This may be due to 281282 binding to a region that is not amenable to HDX-MS analysis". This is an example where it is highly likely that the changes in deuterium uptake are outside the time window one is monitoring especially because other assays indicate that antibody 281282 should have a similar binding affinity as 251249 and 252248. But this study uses only 1 time point (i.e, 5 mins), which is unable to capture the dynamics of binding. I suggest that the authors rewrite this sentence to reflect such a possibility.

(c) On page#16, the sentence "Antibodies 250233, 267268, 277278 and 283284, which showed no inhibition in the reticulocyte-binding assay (Fig. 2A), have partially overlapping epitopes with 237235 and 243244 but showed no changes in the region between Y186-I199 by HDX-MS (Fig. 4)." is another example of where the antibodies 250233, 267268, 277278 and 283284 showed no HDX activity because they may be slow exchanging regions. Moreover the antibody 283284 does show activity close to the Y186-I199 region and Tfr1 region.

2) On page#16, the 326327 antibody is reported as the only mAb which is inhibitory in the reticulocyte-binding assay and IP assay, but not the FRET-based assay. This mAb shares epitopes with 10B12, 256257, 260261 and 346343, and competes for protein binding with 10B12 (Fig. 3C), as shown by the authors. Further, the authors conclude from previous SAXS solution structure data of the PvRBP2b-10B12 complex along with HDX data that antibody 326327 does not bind to the Tfr1-Tf interface. Further they provide an explanation of how steric hindrance may be occurring, which is acceptable. But Fig. 4A shows that 326327 and 10B12 bind at the Tf (417-432) region.

Also, I am curious and am hoping that the authors would also comment on their result that shows mAb 326327 sharing epitopes with antibody 6H1 which is near the Tf region in Fig. 4A and 326327 also shows activity in the 405-423 region which is near to the Tfr1-Tf interaction region (Fig 4Bvii). Of course it would help to have crystallography data for antibody 326327 to clarify things. But is it possible that the binding of 326327 is near the Tfr1-Tf region if not directly on it?

3) I am curious if the authors can get some more insights about mAb 237235, from the X-ray crystal structures of 3E9 binding to PvRBP2b, because 3E9 shares epitope with 237235 according to their HDX results. Can they use that info to validate their findings about mAb 237235 which is proposed to allosterically affect binding of the PvRBP2b to Tf?

Minor corrections:

1) Different parts of the manuscript report different % of sequence coverage, e.g., 85% on page #10,

while 90 % on page 33 (under HDX methods), and 95% on Fig. S6. Please correct this inconsistency.

2) Also, I suggest that the authors make the numbering of the sequence in Fig. S6 consistent with that used in the main text; e.g., on page#10 the authors write "...with limited information for regions 234-322 and 589-686 (Supplementary Fig. S6)." Here the regions 234-322 and 589-686 do not match the same regions in Fig. S6, thus in its current form it is confusing for the readers; for they have to go back and forth between the numbering of the sequence in Fig. S6 and the main text and figure it out themselves.

3) In Fig. S6, the authors should clarify what they mean by the abbreviation cs2, cs3, etc. I think that they are referring to a charge state and thus "cs", however it is best to define the abbreviation.

4) As for Tables S2 and S3, I strongly recommend that the authors transport this data into Excel spreadsheets so that it can be followed easily by the readers. It is very difficult to follow the data in its current format, i.e. in PDF. Reporting in Excel is something that is highly recommended by the HDX community as well, shown here <https://www.nature.com/articles/s41592-019-0459-y>. In fact authors can use the Tables shown in this review article as a template for their HDX data.

5) Also, in Tables S2 and S3, as I understand the heading of the 5th column should be "Sequence" and not "Start"? Please make the change accordingly.

6) In Fig. 1D (in the 2D iso-affinity plot), I suggest that the authors use different symbols to show the different mAb data. Currently it is confusing because all data symbols are the same – open circles.

7) On page#32, under HDX Methods, the authors have reported the pH and composition of the quenching buffer and the temperature of the buffering reaction. I suggest that if possible, the authors also report the pH of the quenched sample along with its concentration because that it what matters. This is another good practice as suggested here (<https://www.nature.com/articles/s41592-019-0459-y>).

8) On page # 33 under the HDX methods section, the authors state that "Results are presented as percentage of maximal deuteration using a fully deuterated sample prepared by incubating the protein for 1 h in 1 M Gdn-HCl before incubation for 6 h in deuterated buffer." As suggested in Nature Methods (<https://www.nature.com/articles/s41592-019-0459-y>), generally a "maximally deuterated sample" is one that has undergone 12-24 hours of deuteration. Though there is no hard and fast rule about the duration for complete deuteration, the longer duration the better. I think that the "fully deuterated" that the authors are eluding may not be fully deuterated. Instead the authors should refer to this as the "highly deuterated sample".

REVIEWER COMMENTS

Reviewer #1 (Remarks to the Author):

This paper by Chan et al describes naturally occurring antibodies to *P. vivax* and it is an interesting contribution to the field of knowledge and there is a lot of data in the paper. The isolation and characterization of these antibodies will be important for further studies. However, there are issues that I think needs some clarification.

1) The authors have performed both ammonium thiocyanate assays and bio-layer interferometry for measurement of avidity, but I can't really see any correlation analysis comparing with the rest of the data. So, does it make a difference whether the antibodies bind with high or low avidity? And which of the two methods (if any) match the other results?

Response: All high avidity antibodies showed picomolar affinities except 237235, which showed nanomolar affinity. However, there is no correlation between the moderate and low avidity antibodies with antibody affinity measurements. With regards to the other results, only 346343, an inhibitory antibody, showed high avidity with picomolar affinity. Otherwise, there was no correlation between inhibitory activity, binding affinities and avidity for this collection of human monoclonal antibodies.

We have added the following sentence to summarize the avidity and affinity measurements to the results section titled '**Isolation and characterization of naturally acquired human mAbs to PvRBP2b**', line 107:

“All high avidity antibodies showed picomolar affinities except 237235, which showed low nanomolar affinity. There was no correlation in the moderate and low avidity antibodies to the affinity measurements.”

2) When looking at the data in supplementary fig S2, all the K_a values look very similar but K_d values differ a lot between high and low values. If an antibody binds with a good K_d , it should be able to stay bound to its target for a longer time period and hence be more effective in its function?

Response: In terms of inhibitory antibodies, the most important aspect is whether the human antibody binds to a site on PvRBP2b that inhibits its interaction with TfR1. Therefore, while an antibody can have good K_d or K_a values, it may not inhibit if it binds outside of the TfR1 receptor engagement site on PvRBP2b.

3) Transferrin is a protein that circulates in plasma. What about binding of the antibodies to free transferrin, compared to in complex with TfR1? If all antibodies are bound to free Transferrin, perhaps none will bind to the complex?

Response: The human antibodies described in this manuscript are specific for the parasite ligand PvRBP2b and therefore are not expected to bind transferrin. Furthermore, in Figure 2c, using anti-PvRBP2b inhibitory human antibodies (239229, 241242, 253245, 258259, 260261, 262231, 273264, 346343), we observe immuno-precipitation of recombinant PvRBP2b but not of transferrin or TfR1, showing that these antibodies are specific to PvRBP2b.

4) Something that is not discussed in this paper is the presence of iron. The affinity of TfR1 for Transferrin changes in the presence of iron and there is also a pH-dependent cooperativity of lobes of transferrin. This might have a big impact on binding of antibodies. How much iron is present in the different assays? Is the pH the same in all assays? The issue of iron is my major concern, since the whole system of TfR1 and transferrin is built up for taking care of iron and in a human body in real life it could have a major impact on binding of antibodies.

Response: The human antibodies described in this manuscript are specific for the parasite ligand PvRBP2b and as such, they are not expected to bind to either transferrin or TfR1. Therefore, the binding of the human antibodies to PvRBP2b will not be affected whether or not transferrin is in the presence of iron. On a separate aspect for PvRBP2b engagement with TfR1 and transferrin (if the reviewer is interested), we have shown previously (Gruszczyk et al., Science, 2018) that PvRBP2b binds similarly to TfR1 either in complex with iron-loaded or iron-depleted Tf.

Reviewer #2 (Remarks to the Author):

In the manuscript “Naturally acquired blocking human monoclonal antibodies to Plasmodium vivax reticulocyte binding protein 2b”, Chan et al. characterize a set of naturally acquired human antibodies against a blood-stage antigen of *P. vivax*. The authors use a broad array of biochemical techniques to identify eight PvRBP2b human mAbs that inhibit the binding of PvRBP2b to reticulocytes by blocking complex formation with TfR1 and Tf. A

nice trend is established whereas mAbs that compete with each other in ELISA assays share similar phenotypes in the functional assays. The study is further substantiated by extensive molecular insights obtained by HDX-MS and X-ray crystallography. Overall this is an impactful report and comprehensive study that describes the molecular basis of P.vivax inhibition by human monoclonal antibodies and suggests possible use of mAbs as therapeutics for P. vivax malaria. The manuscript will be of interest to the broad journal readership once the concerns below are addressed:

1) Fig. 1A: VH genes should be labeled in the panel, at least for the most frequent ones already highlighted in color.

Response: As requested by the reviewer, we have labeled the VH genes for the most frequent antibodies highlighted in color in Fig. 1A. The VH genes are listed in Supplemental Table 1 is consistent with the colors in the pie chart. We have added the following sentence to the **Figure 1a** legend to make this clear:

“VII genes for clonal groups that are colored are also shown in Supplementary Table S1.”

2) The statement “Since antibodies from the same clonal group recognize the same epitope [...]” is unsubstantiated.

Response: We agree with the reviewer and this sentence on line 75 has been changed to: “One or two clones were selected for expression from several of the largest clonal groups from both individuals.”

3) The authors conduct binding affinity and binding avidity measurements, as described in Fig. 1D and 1E. Because the antibodies are not labeled in Fig. 1D, it is impossible to know whether the high-avidity antibodies are also the high-affinity antibodies. It would be helpful if the antibodies could be labeled in Fig. 1D to allow this determination by visual inspection, or perhaps even better if an affinity vs avidity association plot could be added to Fig. 1.

Response: As requested by the reviewer, Fig. 1D antibodies have now been labelled and the actual values for K_D can be found in Supplementary Fig. S2. **Fig. 1D** legend has been modified as follows:

“Iso-affinity plot showing the range of dissociation rate constants (k_d) and association rate constants (k_a) of human mAbs as measured by bio-layer interferometry. Symbols that fall on the same diagonal dotted lines have the same equilibrium dissociation rate constants (K_D) indicated on the top and right sides of the plot. Numeric values for affinity measurements are shown in Supplementary Fig. S2 for each human mAb.”

We have also added the following text to line 107 in the results section titled ‘**Isolation and characterization of naturally acquired human mAbs to PvRBP2b**’:

“All high avidity antibodies showed picomolar affinities except 237235, which showed nanomolar affinity. There was no correlation in the moderate and low avidity antibodies to affinity measurements.”

4) In Fig. 3B, it is unclear how the dark black boxes presumably delineating different epitope bins were defined. This information should be provided in the figure legend. To this Reviewer, it is unclear that those epitope bin boxes were drawn in the most effective way. For example, couldn't the large top left box be subdivided into three more precise sub-boxes?

Also, a singular box is drawn at the bottom right around mAbs 250233 and 267268. But is there any evidence that these two mAbs actually compete with one another and constitute an epitope bin?

Response: The original boxes in Fig. 3B are defined from Fig. 3A, which reflects the amino acid region bound by these antibodies and do not indicate epitope bins. We agree with the reviewer that the boxes are not drawn in the most effective way and therefore these boxes have been removed to avoid confusion. In addition, outlining epitope bins is not possible as several antibodies including 253245, 260261 and 346343 compete with different groups of antibodies.

5) In the Discussion, the authors propose that three clusters of their inhibitory antibodies may be affected by field polymorphisms found in PvRBP2b. Please highlight the locations of the documented polymorphisms in the figure of the PvRBP2b:Fab complex structures. Additionally, a supplemental figure showing the sequence of PvRBP2b with the polymorphic regions and mAbs binding regions highlighted would be a helpful resource for the reader.

Response: As requested, we have provided a new Supplementary Fig. S9 that highlights the position of documented polymorphisms on the sequence of the TfR1 -Tf binding domain of PvRBP2b (Fig. S9A) and labeled these polymorphisms in the context of the PvRBP2b:Fab crystal structures (Fig. S9B). We have not highlighted the mAb binding regions within the sequence as this is difficult to do with an alpha helical structure where antibodies contact only one side of the helix. In the discussion, we have decided to discuss polymorphisms that may affect antibody binding as inferred by the crystal structures, as the HDX-MS data does not give precise epitopes. In the discussion, we have modified to the following paragraph as below (line 416):

“One of the major challenges in targeting malaria proteins for vaccination are the presence of naturally occurring polymorphisms that may limit the effective use of a vaccine (reviewed in ²¹). The N-terminal domain of PvRBP2b encompassing residues S168-E460 is under balancing selection ^{11,12}. Targeting multiple PvRBP2b epitopes using a combination of human mAbs may lessen the impact of field polymorphisms that may disrupt antibody binding. Of the four sets of antibodies that may be useful in combination, structural analyses reveal two field polymorphisms that may affect 241242/273264 binding. 241242/273264 binds in the N-terminal domain where seven polymorphic residues occur and the crystal structures reveal their binding may be affected by K363 and G382 polymorphisms (Supplementary Fig. S9). As the precise epitopes of 239229/262231 and 326327 are unknown, we cannot predict how polymorphisms may affect their binding. Further mutational studies will be needed to confirm whether these human mAbs are affected by existing field polymorphisms. Of the four antibody sets, only 237235 binds a highly conserved region within the N-terminal domain. The only polymorphism within close proximity is R242, which is located directly adjacent to the 237235 epitope (Supplementary Fig. S9). In future studies, it would be important to isolate and examine more human mAbs that bind within the C-terminal region, which has only four polymorphisms and in particular no polymorphisms that reside in the TfR1 interaction site.”

6) Another question relating to the naturally occurring polymorphisms: was the PvRBP2b protein construct used in the study typical of a circulating East-Asian *P. vivax* isolate? A lot of questions regarding the versatility of selected antibodies could be answered by performing

bio-layer interferometry experiments with PvRBP2b from geographically distant *P. vivax* isolates (from Brazil, etc), if these recombinant proteins are already available.

Response: Recombinant PvRBP2b was derived from the *Salvador I P. vivax* isolate. Unfortunately, we do not have the recombinant proteins available for bio-layer interferometry experiments from geographically distant *P. vivax* isolates. As *P. vivax* harbors five-fold more genetic diversity than *P. falciparum*, to generate the numerous polymorphic recombinant proteins would be an extensive task and beyond the scope of this manuscript.

7) In the Discussion, the authors suggest that “241242/273264, 239229/262231, 326327 and 237235 could be examined together in *P. vivax* ex vivo invasion assays to ascertain whether a combination of antibodies is more effective than using single human mAbs for inhibiting parasite invasion.” This proposition is attractive; but it is unclear why this experiment wasn’t attempted. The impact of the reported biophysical results and overall conclusion of the manuscript would be considerably strengthened should this investigation be completed.

Response: We agree that these are important experiments, but *ex vivo* invasion assays for *P. vivax* are extremely difficult and almost impossible during the COVID-19 pandemic. We feel that it is important to publish the considerable data we already have and to make our antibodies readily available to other research groups in malaria endemic regions to perform an in-depth study on how these antibodies perform in inhibiting different geographical strains of *P. vivax*.

8) 279280 binding to PvRBP2b displays signs of antibody-dependent enhancement in FRET assays and flow cytometry-based reticulocyte binding. Can the authors speculate what might be the molecular basis of this phenomenon?

Response: By HDX-MS 279280 upon binding shows changes in PvRBP2b opposite the TfR1 binding site. Therefore 279280 could alter the region where TfR1 binds to make it favorable for TfR1 interaction. We have included the following sentence on line 380 in the discussion to address this question:

“279280 enhances PvRBP2b binding to TfR1-Tf in both the reticulocyte binding assay and FRET assay. By HDX-MS, 279280 binds PvRBP2b in a region directly opposite the TfR1 binding site and it could stabilize this region to make it favorable for TfR1 binding. These epitopes that elicit enhancing antibodies are important to elucidate as they would be detrimental in a vaccine antigen.”

9) In the second molecule of the ASU for the PvRBP2b:241242 crystal structure, why is so little of the Fab constant domain for chains E and F built (PDB ID: 6WN1)? The R_{work}/R_{free} may improve slightly if the remaining model is built into any visible density.

Response: Chains E and F of the second molecule of 241242 in the PvRBP2b:241242 crystal structure (PDB ID:6WN1) did not show enough electron density to confidently build in the Fab constant domain. Deleting these regions improved the R_{work}/R_{free} values.

0) In Supplementary Table 5 (crystallography table), the authors should report all values to the adequate significant figure, and be consistent across all entries of the same type.

Response: We thank the reviewer for the comment and the crystallography table in Supplementary Table 6 has been revised.

Reviewer #3 (Remarks to the Author):

The manuscript titled “Naturally acquired blocking human monoclonal antibodies to Plasmodium vivax reticulocyte binding protein 2b” is authored by Chan et al. Individuals with higher levels of antibodies to PvRBP2b have been shown to have reduced risk of P. vivax infection and disease, however the reason underlying this is still unknown. In this manuscript, the authors report the isolation and characterization of naturally acquired anti-PvRBP2b human monoclonal antibodies (mAbs) from two individuals in an effort to determine and understand the structural mechanisms by which they inhibit binding to PvRBP2. The authors have used an array of biophysical assays, e.g, BLI, FRET, to determine which mAbs bind and their binding affinities to PvRBP2. Coupled with these they have also used X-ray crystallography and solution phase hydrogen-deuterium exchange (HDX), to map the binding sites/epitopes on PvRBP2.

The idea seems fundamentally sound to me, however I am not as familiar with the biology and thus feel most comfortable commenting on the HDX methods and analysis part of the manuscript. Overall I think that the manuscript is well written, the research question is clear and well-addressed.

I also think that this study will be of interest to the HDX community because solution-phase HDX is increasingly becoming a tool of choice to probe protein-antibody interactions and map epitopes in order to provide understanding of the underlying basis of such interactions, and is also being implemented in the biopharma industry. I recommend this manuscript for publications after addressing the issues below. Once the concerns below have been properly addressed, this could be a manuscript worthy of publication in Nature Communications.

HDX was performed to map put the epitopes of for PvRBP2-mAbs, by comparing the deuterium uptake data in the presence and absence of the mAbs.

Major issues:

1) HDX methods: I understand that the authors isolated many human mAbs and thus have had to work with many mAbs. So I appreciate the time and effort they put in performing HDX experiments with these many mAbs. However it would benefit the readers if the authors justified their selection of 1 time points (5 mins). In fact choosing more time points and a longer time point than 5 mins, would have benefitted the authors. I am in no way suggesting that the authors redo the experiment, however one big caveat of having one time point is that if a region does change in the bound state, the change might not take place at the time point chosen (i.e. 5 mins) for the experiment. For e.g., if a protein that binds an antibody has 7 regions that change, HDX at one single time point may pick up only 4 out of those 7 regions because the other 3 happen are either at a fast or a slow time scale. So those 3 regions may appear as they have not changed. Thus no change actually means that there is no observed change in HDX, because the change happened outside of the time frame of the experiment.

Response: We are very grateful to this comment regarding the choice of incubation times in D2O and fully aware that we can miss changes because we can simply not capture them with the chosen timepoints of 5 min. The suggestion to justify how the incubation times were chosen is very important and we thank the reviewer for suggesting to include this in the manuscript.

The conditions were carefully chosen to maximize our chances of seeing changes in a maximal number of regions of RBP2b. As mentioned by the reviewer, the fact that we were

screening so many antibodies forced us to choose few conditions that offered the maximal chances of capturing HDX changes in presence of antibodies. By doing 5 minutes incubations at either 0°C or 22°C, we observe very different exchange rates that can be accounted for two different incubation conditions.

To address this point, we had first tested multiple incubation times, at 0°C for 3sec, 30sec, 5min and 50min and at 22°C for 5min and 50min. We then compared the D2O incorporation of each peptides with the highly deuterated sample that was incubated for 6h at 22°C in presence of 1M Gdn-HCl. We also tested most of those conditions for the mouse antibodies 3E9, 6H1 and 10B12 to see in which conditions changes in deuteration levels could be observed.

These experiments were very conclusive and showed that:

- 1) When incubating for 30sec or less at 0°C, a lot of peptides showed a minimal D2O incorporation. With this incubation times, changes occurring in regions with minimal HDX rates could easily be missed.
- 2) Incubation for 50min at 22°C is almost equivalent to the highly deuterated sample. With this in mind, we feared that too many peptides would be maximally deuterated and may not show a change upon mAb binding.
- 3) Incubation for 5min at 0°C and at 22°C were chosen because many peptides showed intermediate D2O incorporation in those conditions. Moreover, D2O incorporation was distinct enough in those two conditions to capture changes in peptides showing either low or high HDX rates.
- 4) Changes in deuteration levels induced by mouse antibodies were visible at all tested timepoints but were less intense at 3 sec incubation at 0 °C.

(a) With this in mind, I suggest that the authors re-write the sentence on Page#10 “Protein amide protons were labelled with deuterium for five min at either 0 °C or 22°C in deuterated buffer, providing information on the fast-exchanging and slow-exchanging regions of the protein.” Because there will be way slower exchanging regions which will not begin exchanging in 5 minutes, & may take up to several hours.

Response: This sentence on line 210 was replaced in the results section titled ‘**Hydrogen deuterium exchange epitope mapping**’ with the following one to clarify the fact that changes in most regions of the protein should be observable by HDX-MS:

“To identify the best HDX conditions that would allow us to screen all antibodies, we first tested different incubation times for PvRBP2b_{161–1454} alone both at 0 °C and at room temperature (22 °C) (Table S2). These experiments showed that incubation times of 5 min at either 0 °C or 22 °C were the most appropriate as they showed intermediate levels of deuterium incorporation for a majority of peptides, both for the fast-exchanging and slow-exchanging regions of the protein.”

(b) On page #11 where the authors write that “Antibody 281282 likely binds very similarly to 251249 and 252248 as all three antibodies have the same profile in the competition assay (Fig. 3B), but no HDX changes were observed in the presence of antibody 281282 (Fig. 4A). This may be due to 281282 binding to a region that is not amenable to HDX-MS analysis”. This is an example where it is highly likely that the changes in deuterium uptake are outside the time window one is monitoring especially because other assays indicate that antibody 281282 should have a similar binding affinity as 251249 and 252248. But this study uses only 1 time point (i.e, 5 mins), which is unable to capture the dynamics of binding. I suggest that the authors rewrite this sentence to reflect such a possibility.

Response: The sentence on line 237 was rewritten to better explain the reasons why no changes are observed:

“This may be because the chosen HDX incubation time and temperatures do not permit observable changes in the region where 281282 is binding, or because the antibody is contacting a region that is not covered in the HDX-MS analysis.”

(c) On page#16, the sentence “Antibodies 250233, 267268, 277278 and 283284, which showed no inhibition in the reticulocyte-binding assay (Fig. 2A), have partially overlapping epitopes with 237235 and 243244 but showed no changes in the region between Y186-I199 by HDX-MS (Fig. 4).” is another example of where the antibodies 250233, 267268, 277278 and 283284 showed no HDX activity because they may be slow exchanging regions. Moreover the antibody 283284 does show activity close to the Y186-I199 region and TfR1 region.

Response: We agree with reviewer 3 comment. This issue is addressed in the previous correction.

2) On page#16, the 326327 antibody is reported as the only mAb which is inhibitory in the reticulocyte-binding assay and IP assay, but not the FRET-based assay. This mAb shares epitopes with 10B12, 256257, 260261 and 346343, and competes for protein binding with 10B12 (Fig. 3C), as shown by the authors. Further, the authors conclude from previous SAXS solution structure data of the PvRBP2b-10B12 complex along with HDX data that antibody 326327 does not bind to the TfR1-Tf interface. Further they provide an explanation of how steric hindrance may be occurring, which is acceptable. But Fig. 4A shows that 326327 and 10B12 bind at the Tf (417-432) region.

Response: The HDX change induced by 326327 at the 405-423 region, which overlaps with the Tf binding region (417-432) may well be induced by the binding of 326327 to the loop at the extremity of PvRBP2b for which we do not have coverage (308-322). This would modify the conformation of adjacent regions 405-423 and 212-232. As mentioned by the reviewer, there may not be a direct contact between 326327 and Tf even if we observe HDX reductions upon 326327 binding that correlate with Tf binding regions.

As we could not validate this hypothesis experimentally, we decided not to discuss details of this HDX-MS data in the text. They do not modify our view of the inhibitory mode of antibody 326327. Furthermore, we believe that commenting details of the change seen with 326327 would complicate the discussion rather than clarify the mode of action.

Also, I am curious and am hoping that the authors would also comment on their result that shows mAb 326327 sharing epitopes with antibody 6H1 which is near the Tf region in Fig. 4A and 326327 also shows activity in the 405-423 region which is near to the TfR1-Tf interaction region (Fig 4Bvii). Of course it would help to have crystallography data for antibody 326327 to clarify things. But is it possible that the binding of 326327 is near the TfR1-Tf region if not directly on it?

Response: We have added the following sentence to line 370 to extend the discussion on mAb 326327 mode of action:

“326327 binding also likely triggers allosteric changes in regions where other antibodies bind (6H1, 256257, 260261 and 346343).”

3) I am curious if the authors can get some more insights about mAb 237235, from the X-ray crystal structures of 3E9 binding to PvRBP2b, because 3E9 shares epitope with 237235 according to their HDX results. Can they use that info to validate their findings about mAb 237235 which is proposed to allosterically affect binding of the PvRBP2b to Tf?

Response: From previously determined crystal structures of 3E9 binding to PvRBP2b (Gruszczyk and Huang et al., Nature, 2018), its mode of inhibition is by direct steric hindrance with TfR1, so whether allosteric changes upon 3E9 binding also causes inhibition cannot be investigated.

Minor corrections:

1) Different parts of the manuscript report different % of sequence coverage, e.g., 85% on page #10, while 90 % on page 33 (under HDX methods), and 95% on Fig. S6. Please correct this inconsistency.

Response: Protein coverage level was corrected to the final 85% everywhere.

2) Also, I suggest that the authors make the numbering of the sequence in Fig. S6 consistent with that used in the main text; e.g., on page #10 the authors write “...with limited information for regions 234-322 and 589-686 (Supplementary Fig. S6).” Here the regions 234-322 and 589-686 do not match the same regions in Fig. S6, thus in its current form it is confusing for the readers; for they have to go back and forth between the numbering of the sequence in Fig. S6 and the main text and figure it out themselves.

Response: This is a good point. We have corrected the numbering of the protein to match the endogenous sequence in Fig. S6.

3) In Fig. S6, the authors should clarify what they mean by the abbreviation cs2, cs3, etc. I think that they are referring to a charge state and thus “cs”, however it is best to define the abbreviation.

Response: Description of the abbreviation was added in Figure Legend of Fig. S7.

4) As for Tables S2 and S3, I strongly recommend that the authors transport this data into Excel spreadsheets so that it can be followed easily by the readers. It is very difficult to follow the data in its current format, i.e. in PDF. Reporting in Excel is something that is highly recommended by the HDX community as well, shown here <https://www.nature.com/articles/s41592-019-0459-y>. In fact authors can use the Tables shown in this review article as a template for their HDX data.

Response: Tables S2 and S3 have been resubmitted as excel spreadsheets.

5) Also, in Tables S2 and S3, as I understand the heading of the 5th column should be “Sequence” and not “Start”? Please make the change accordingly.

Response: Column heading was modified to “Sequence”.

6) In Fig. 1D (in the 2D iso-affinity plot), I suggest that the authors use different symbols to

show the different mAb data. Currently it is confusing because all data symbols are the same – open circles.

Response: Fig. 1D antibodies have now been labelled using different symbols and actual numbers for K_D can be found in Supplementary Fig. S2. **Fig. 1D** legend has been modified and now reads:

“(D) Iso-affinity plot showing the range of dissociation rate constants (k_d) and association rate constants (k_a) of human mAbs as measured by bio-layer interferometry. Symbols that fall on the same diagonal dotted lines have the same equilibrium dissociation rate constants (K_D) indicated on the top and right sides of the plot. Numeric values for affinity measurements are shown in Supplementary Fig. S2 for each human mAb.”

7) On page#32, under HDX Methods, the authors have reported the pH and composition of the quenching buffer and the temperature of the buffering reaction. I suggest that if possible, the authors also report the pH of the quenched sample along with its concentration because that it what matters. This is another good practice as suggested here (<https://www.nature.com/articles/s41592-019-0459-y>).

Response: The pH and protein amount are critical values for HDX-MS experiments. In our manuscript we provide all details about the protein buffer, the deuterated buffer and the quench buffer used. The volumes used are also described. This allows any scientist to repeat the experiment in the exact same conditions as we have performed. For this reason, we believe that the final pH of the quenched sample does not need to be measured. Moreover, it is difficult to assess the pH of a 70 μ l sample.

8) On page # 33 under the HDX methods section, the authors state that “Results are presented as percentage of maximal deuteration using a fully deuterated sample prepared by incubating the protein for 1 h in 1 M Gdn-HCl before incubation for 6 h in deuterated buffer.” As suggested in Nature Methods (<https://www.nature.com/articles/s41592-019-0459-y>), generally a “maximally deuterated sample” is one that has undergone 12-24 hours of deuteration. Though there is no hard and fast rule about the duration for complete deuteration, the longer duration the better. I think that the “fully deuterated” that the authors are eluding may not be fully deuterated. Instead the authors should refer to this as the “highly deuterated sample”.

Response: The text was modified in line 772 to refer to “highly deuterated sample”.

Reviewers' Comments:

Reviewer #1:

Remarks to the Author:

I am satisfied with the additions and explanations made to the manuscript.

Reviewer #2:

Remarks to the Author:

The authors have adequately addressed all concerns raised during the initial review of this important manuscript. Only two minor points remain:

1) The isolate from which the recombinant PvRBP2b protein constructs derive (Salvador I *P. vivax* isolate) should be listed in the Materials and Methods. How the use of this isolate instead of an East-Asian isolate may impact the characterization of mAbs derived from East-Asian samples should also be discussed. Indeed, as the authors state in their rebuttal: "*P. vivax* harbors five-fold more genetic diversity than *P. falciparum*".

2) Perhaps the resolution stated for structure 6WTY (Supp Table 6) should be revised. Indeed, all other structures reported in this manuscript (i.e. 6WM9, 6WN1, 6WNO and 6WOZ) have reasonable I/sigma values between 1.3 and 1.8, and reasonable CC1/2 values between 72.4% and 57.2% in the highest resolution shells. In contrast, 6WTY has extremely low values of 0.5 for I/sigma and 13.1% for CC1/2. It is this Reviewer's opinion that the resolution of 6WTY should be adjusted to be in line with the I/sigma and CC1/2 values of the other structures.

Reviewer #3:

Remarks to the Author:

The authors have satisfactorily responded to all my questions & the revised version of the manuscript appears to be good. And I recommend publication.

REVIEWER COMMENTS

Reviewer #1 (Remarks to the Author):

I am satisfied with the additions and explanations made to the manuscript.

Response: Thank you!

Reviewer #2 (Remarks to the Author):

The authors have adequately addressed all concerns raised during the initial review of this important manuscript. Only two minor points remain:

1) The isolate from which the recombinant PvRBP2b protein constructs derive (Salvador I *P. vivax* isolate) should be listed in the Materials and Methods. How the use of this isolate instead of an East-Asian isolate may impact the characterization of mAbs derived from East-Asian samples should also be discussed. Indeed, as the authors state in their rebuttal: "*P. vivax* harbors five-fold more genetic diversity than *P. falciparum*".

Response: We have now indicated in the Materials and Methods that our protein sequences are derived from the *Salvador I P. vivax* isolate, line 631:

"The constructs were derived from the *P. vivax* strain *Salvador I*, obtained from PlasmoDB Database (www.plasmodb.org; accession number: PVX_094255)".

We have added the following sentence to the discussion to highlight the importance of looking at these antibodies in Southeast Asian isolates of *P. vivax* where polymorphisms may affect antibody function, line 418:

“Our human mAbs were characterized using PvRBP2b sequences from the reference *P. vivax* strain *Salvador I* isolated from El Salvador and it is therefore important to investigate the action of these antibodies in Southeast Asian strains of *P. vivax* where our antibodies originate and where polymorphisms may affect antibody function.”

2) Perhaps the resolution stated for structure 6WTY (Supp Table 6) should be revised. Indeed, all other structures reported in this manuscript (i.e. 6WM9, 6WN1, 6WNO and 6WOZ) have reasonable I/σ values between 1.3 and 1.8, and reasonable $CC_{1/2}$ values between 72.4% and 57.2% in the highest resolution shells. In contrast, 6WTY has extremely low values of 0.5 for I/σ and 13.1% for $CC_{1/2}$. It is this Reviewer's opinion that the resolution of 6WTY should be adjusted to be in line with the I/σ and $CC_{1/2}$ values of the other structures.

Response: Whereas the initial, more conservative, resolution cut-offs proved sufficient for molecular replacement, we found that better ($2mF_{obs} - DF_{calc}$, a_{calc}) electron density maps could be obtained by extending the resolution range of the diffraction data employed to the maximum judged statistically significant by the XDS package. Model building into these maps also succeeding in reducing the subsequent difference between R_{free} and R_{work} from 0.10 to 0.04. The ability of weak data inclusion to improve crystallographic refinement at low resolution is well documented (see, for example, Diederichs K, and Karplus PA. Better models by discarding data? *Acta Crystallographica Section D* (2013) 69:1215-22, and Evans PR, and Murshudov GN. How good are my data and what is the resolution? *Acta Crystallographica Section D* (2013) 69:1204-14.). Nevertheless, we have altered the wording within the methodology to make this point clear, line 837:

“To improve the refinement, the resolution cut-off for diffraction data for 253245 was extended to 3.48 Å to make maximum use of weak data in the outer shell (3.69 - 3.48 Å)—within this outer resolution shell, $CC_{1/2}$ remained significant at the $P = 0.001$ level and data completeness remained above >90%.”

Reviewer #3 (Remarks to the Author):

The authors have satisfactorily responded to all my questions & the revised version of the manuscript appears to be good. And I recommend publication.

Response: Thank you!